# DENOISING MCMC FOR ACCELERATING DIFFUSION-BASED GENERATIVE MODELS

## ABSTRACT

Diffusion models are powerful generative models that simulate the reverse of diffusion processes using score functions to synthesize data from noise. The sampling process of diffusion models can be interpreted as solving the reverse stochastic differential equation (SDE) or the ordinary differential equation (ODE) of the diffusion process, which often requires up to thousands of discretization steps to generate a single image. This has sparked a great interest in developing efficient integration techniques for reverse-S/ODEs. Here, we propose an orthogonal approach to accelerating score-based sampling: Denoising MCMC (DMCMC). DMCMC first uses MCMC to produce initialization points for reverse-S/ODE in the product space of data and variance (or diffusion time). Then, a reverse-S/ODE integrator is used to denoise the initialization points. Since MCMC traverses close to the data manifold, the cost of producing a clean sample for DMCMC is much less than that of producing a clean sample from noise. To verify the proposed concept, we show that Denoising Langevin Gibbs (DLG), an instance of DMCMC, successfully accelerates all six reverse-S/ODE integrators considered in this work on the tasks of CIFAR10 and CelebA-HQ-256 image generation. Notably, combined with integrators of Karras et al. (2022) and pre-trained score models of Song et al. (2021b), DLG achieves state-of-the-art results among score-based models. In the limited number of score function evaluation (NFE) settings on CIFAR10, we have 3.86 FID with $\approx$ 10 NFE and 2.63 FID with $\approx$ 20 NFE. On CelebA-HQ-256, we have 6.99 FID with $\approx$ 160 NFE, which beats the current best record of Kim et al. (2022) among score-based models, 7.16 FID with 4000 NFE.

## 1 INTRODUCTION

Sampling from a probability distribution given its score function, i.e., the gradient of the log-density, is an active area of research in machine learning. Its applications range far and wide, from Bayesian learning (Welling & Teh, 2011) to learning energy-based models (Song & Kingma, 2021), synthesizing new high-quality data (Dhariwal & Nichol, 2021), and so on. Typical examples of traditional score-based samplers are Markov chain Monte Carlo (MCMC) methods such as Langevin dynamics (Langevin, 1908) and Hamiltonian Monte Carlo (Neal, 2011).

Recent developments in score matching with deep neural networks (DNNs) have made it possible to estimate scores of high-dimensional distributions such as those of natural images (Song & Ermon, 2019). However, natural data distributions are often sharp and multi-modal, rendering naïve application of traditional MCMC methods impractical. Specifically, MCMC methods tend to skip over or get stuck at local high-density modes, producing biased samples (Levy et al., 2018).

Diffusion models (Sohl-Dickstein et al., 2015; Ho et al., 2020; Song et al., 2021a) depart from MCMC and use the concept of diffusion, the process of gradually corrupting data into noise, to generate samples. Song et al. (2021b) observed that for each diffusion process, there is a reverse stochastic differential equation (SDE) and an ordinary differential equation (ODE). Hence, given a noise sample, integrating the reverse-S/ODE produces a data sample. Only a time-dependent score function of the data during the diffusion process is required to simulate the reverse process.

This discovery generated great interest in finding better ways to integrate reverse-S/ODEs. For instance, Song et al. (2021b) uses black-box ODE solvers with adaptive stepsizes to accelerate sampling. Furthermore, multitude of recent works on score-based generative modeling focus on improv-

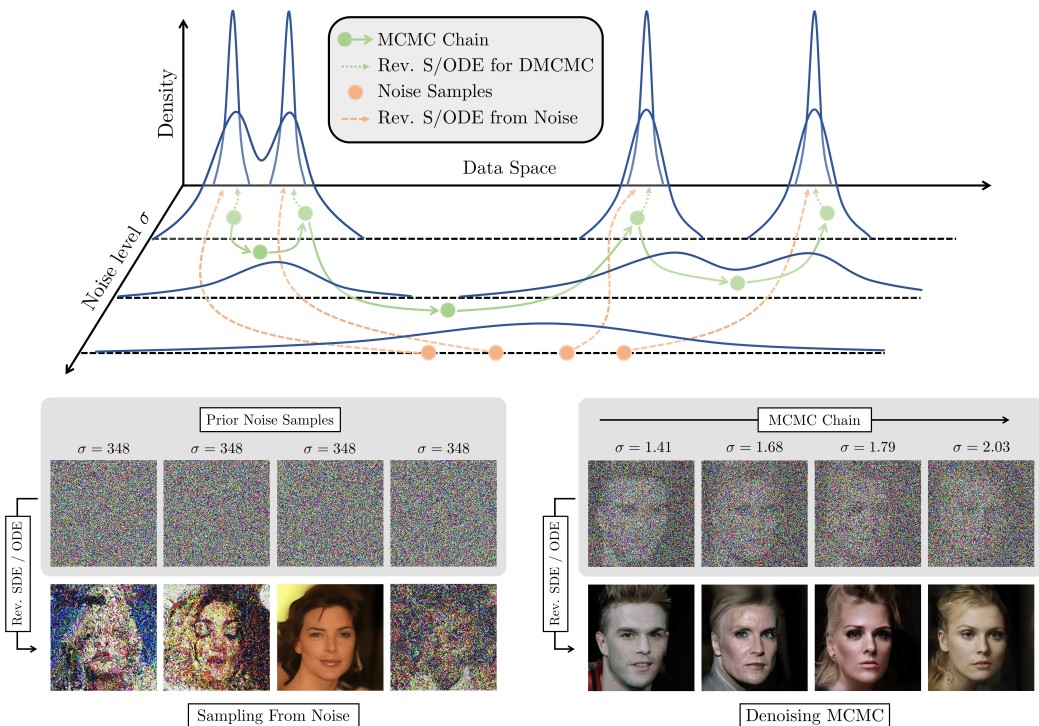

Figure 1: **Top:** a conceptual illustration of a VE diffusion model sampling process and DMCMC sampling process. VE diffusion models integrate the reverse-S/ODE starting from maximum diffusion time / maximum noise level. So, samples are often noisy with small computation budget due to large truncation error. DMCMC produces an MCMC chain which travels close to the image manifold (compare the noise level $\sigma$). So, the MCMC samples can be denoised to produce high-quality data with relatively little computation. **Bottom:** Visualization of sampling processes without (left) and with (right) DMCMC on CelebA-HQ-256 under a fixed computation budget.

ing reverse-S/ODE integrators (Jolicoeur-Martineau et al., 2021; Lu et al., 2022; Karras et al., 2022; Zhang & Chen, 2022).

In this work, we develop an orthogonal approach to accelerating score-based sampling. Specifically, we propose Denoising MCMC (DMCMC) which combines MCMC with reverse-S/ODE integrators. MCMC is used to generate initialization points $\{(\boldsymbol{x}_n, t_n)\}$ in the product space of data $\boldsymbol{x}$ and variance exploding (VE) diffusion time $t$ / noise level $\sigma$ (see Fig. 1 top panel). Since all modes are connected in the product space, MCMC mixes well. Then, a reverse-S/ODE integrator solves the reverse-S/ODE starting at $\boldsymbol{x}_n$ from time $t = t_n$ to $t = 0$. Since MCMC explores high-density regions, the MCMC chain stays close to the data manifold, so $t_n$ tends to be close to $0$, i.e., noise level tends to be small (see Fig. 1 top and bottom panels). Thus, integrating the reverse-S/ODE from $t = t_n$ to $t = 0$ is much faster than integrating the reverse-S/ODE from maximum time $t = T$ to $t = 0$ starting from noise. This leads to a significant acceleration of the sampling process.

Our contributions can be summarized as follows.

- We introduce the product space of data and diffusion time, and develop a novel score-based sampling framework called Denoising MCMC on the product space. Our framework is general, as any MCMC, any VE process noise-conditional score function, and any reverse-S/ODE integrator can be used in a plug-and-play manner.

- We develop Denoising Langevin Gibbs (DLG), which is an instance of Denoising MCMC that is simple to implement and is scalable. The MCMC part of DLG alternates between a data update step with Langevin dynamics and a noise level prediction step, so all that DLG requires is a pre-trained noise-conditional score network and a noise level classifier.

- We verify the effectiveness of DLG by accelerating six reverse-S/ODE integrators. Notably, combined with the integrators of Karras et al. (2022), DLG achieves state-of-the-art results among score-based models. On CIFAR10 in the limited number of score function evaluation (NFE) setting, we obtain 3.86 FID with $\approx$ 10 NFE and 2.63 FID with $\approx$ 20 NFE. On CelebA-HQ-256, we have 6.99 FID with $\approx$ 160 NFE, which is currently the best result with score-based models. The computation cost of evaluating a noise level classifier is negligible, so we obtain acceleration essentially for free.

## 2 BACKGROUND

### 2.1 DENOISING SCORE MATCHING

Given a distribution $p(\boldsymbol{x})$, a noise level $\sigma$, and a perturbation kernel $p_\sigma(\boldsymbol{x} \mid \tilde{\boldsymbol{x}}) = \mathcal{N}(\boldsymbol{x} \mid \tilde{\boldsymbol{x}}, \sigma^2 \boldsymbol{I})$, solving the denoising score matching objective (Vincent, 2011)

$$\min_\theta \mathbb{E}_{p(\tilde{\boldsymbol{x}})} \mathbb{E}_{p_\sigma(\boldsymbol{x}|\tilde{\boldsymbol{x}})} \left[ \|\boldsymbol{s}_\theta(\boldsymbol{x}) - \nabla_{\boldsymbol{x}} \log p_\sigma(\boldsymbol{x} \mid \tilde{\boldsymbol{x}})\|_2^2 \right] \tag{1}$$

yields a score model $\boldsymbol{s}_\theta(\boldsymbol{x})$ which approximates the score of $\int p_\sigma(\boldsymbol{x} \mid \tilde{\boldsymbol{x}}) p(\tilde{\boldsymbol{x}}) \, d\tilde{\boldsymbol{x}}$. Denoising score matching was then extended to train Noise Conditional Score Networks (NCSNs) $\boldsymbol{s}_\theta(\boldsymbol{x}, \sigma)$ which approximate the score of data smoothed at a general set of noise levels by solving

$$\min_\theta \mathbb{E}_{\lambda(\sigma)} \mathbb{E}_{p(\tilde{\boldsymbol{x}})} \mathbb{E}_{p_\sigma(\boldsymbol{x}|\tilde{\boldsymbol{x}})} \left[ \|\boldsymbol{s}_\theta(\boldsymbol{x}, \sigma) - \nabla_{\boldsymbol{x}} \log p_\sigma(\boldsymbol{x} \mid \tilde{\boldsymbol{x}})\|_2^2 \right] \tag{2}$$

where $\lambda(\sigma)$ can be a discrete or a continuous distribution over $(\sigma_{\min}, \sigma_{\max})$ (Song & Ermon, 2019; Song et al., 2021b). We note $\int p_\sigma(\boldsymbol{x} \mid \tilde{\boldsymbol{x}}) p(\tilde{\boldsymbol{x}}) \, d\tilde{\boldsymbol{x}}$ approaches $p(\boldsymbol{x})$ as $\sigma \to 0$, since the perturbation kernel $p_\sigma(\boldsymbol{x} \mid \tilde{\boldsymbol{x}})$ converges to the Dirac delta function centered at $\boldsymbol{x}$.

### 2.2 MARKOV CHAIN MONTE CARLO (MCMC)

Given an unnormalized version of $p(\boldsymbol{x})$ or the score function $\nabla_{\boldsymbol{x}} \log p(\boldsymbol{x})$, MCMC constructs a Markov chain in the data space whose stationary distribution is $p(\boldsymbol{x})$. An MCMC which uses the unnormalized density is the Metropolis-Hastings MCMC (Metropolis et al., 1953; Hastings, 1970) that builds a Markov chain by sequentially accepting or rejecting proposal distribution samples according to a density ratio. A popular score-based MCMC is Langevin dynamics (Langevin, 1908). Langevin dynamics generates a Markov Chain $\{\boldsymbol{x}_n\}_{n=1}^\infty$ using the iteration

$$\boldsymbol{x}_{n+1} = \boldsymbol{x}_n + (\eta/2) \cdot \nabla_{\boldsymbol{x}} \log p(\boldsymbol{x}_n) + \sqrt{\eta} \cdot \boldsymbol{\epsilon} \tag{3}$$

where $\boldsymbol{\epsilon} \sim \mathcal{N}(\boldsymbol{0}, \boldsymbol{I})$. $\{\boldsymbol{x}_n\}_{n=1}^\infty$ converges to $p(\boldsymbol{x})$ in distribution for an appropriate choice of $\eta$.

To sample from a joint distribution $p(\boldsymbol{x}, \boldsymbol{y})$, we may resort to Gibbs sampling (Geman & Geman, 1984). Given a current Markov chain state $(\boldsymbol{x}_n, \boldsymbol{y}_n)$, Gibbs sampling produces $\boldsymbol{x}_{n+1}$ by sampling from $p(\boldsymbol{x} \mid \boldsymbol{y}_n)$ and $\boldsymbol{y}_{n+1}$ by sampling from $p(\boldsymbol{y} \mid \boldsymbol{x}_{n+1})$. The sampling steps may be replaced with MCMC. Hence, Gibbs sampling is useful when conditional distributions are amenable to MCMC.

**Annealed MCMC.** Despite their diversity, MCMC methods often have difficulty crossing low-density regions in high-dimensional multimodal distributions. For Langevin dynamics, at a low-density region, the score function vanishes in Eq. (3), resulting in a meaningless diffusion. Moreover, natural data often lies on a low-dimensional manifold. Thus, once Langevin dynamics leaves the data manifold, it becomes impossible for Langevin dynamics to find its way back.

One way to remedy this problem is to use annealing, i.e., constructing a sequence of increasingly smooth and wide distributions and running MCMC at different levels of smoothness. As smoothness is increased, disjoint modes merge, so MCMC can cross over to other modes. Annealing has been used to empower various types of MCMC (Geyer & Thompson, 1995; Neal, 2001). In this work, we shall refer to the collection of MCMC that use annealing as annealed MCMC.

An instance of annealed MCMC is annealed Langevin dynamics (ALD) (Song & Ermon, 2019). For a sequence of increasing noise levels $\{\sigma_i\}_{i=1}^N$, Langevin dynamics is sequentially executed with $\int p_{\sigma_i}(\boldsymbol{x} \mid \tilde{\boldsymbol{x}}) p(\tilde{\boldsymbol{x}}) \, d\tilde{\boldsymbol{x}}$ in place of $p(\boldsymbol{x})$ in Eq. (3) for $i = N, N-1, \ldots, 1$. Since $p(\boldsymbol{x})$ smoothed at a large noise level has wide support and connected modes, ALD overcomes the pitfalls of vanilla Langevin dynamics. However, ALD has the drawback that thousands of iterations are required to produce a single batch of samples.

### 2.3 Diffusion Models

**Diffusion models and differential equations.** Diffusion models opened up a new avenue towards fast sampling with score functions via SDEs and ODEs (Song et al., 2021b). Suppose data is distributed in $\mathbb{R}^d$. Given a diffusion process of data sample $\boldsymbol{x}_0 \sim p(\boldsymbol{x})$ into a sample from a simple prior noise distribution, the trajectory of data during diffusion can be described with an Itô SDE

$$d\boldsymbol{x} = \boldsymbol{f}(\boldsymbol{x}, t)\, dt + g(t)\, d\boldsymbol{w} \tag{4}$$

for some drift coefficient $\boldsymbol{f} : \mathbb{R}^d \times [0, T] \to \mathbb{R}^d$, diffusion coefficient $g : [0, T] \to \mathbb{R}$, and Brownian motion $\boldsymbol{w}$. Here, $T$ is the diffusion termination time. With initial condition $\boldsymbol{x}(0) = \boldsymbol{x}_0$, integrating Eq. (4) from time $t = 0$ to $t = T$ produces a sample from the prior distribution.

For each diffusion SDE, there exists a corresponding reverse-SDE:

$$d\boldsymbol{x} = [\boldsymbol{f}(\boldsymbol{x}, t) - g(t)^2 \nabla_{\boldsymbol{x}} \log p_t(\boldsymbol{x})]\, dt + g(t)\, d\bar{\boldsymbol{w}} \tag{5}$$

where $p_t(\boldsymbol{x})$ is the density of $\boldsymbol{x}(t)$ evolving according to Eq. (4) and $\bar{\boldsymbol{w}}$ is a Brownian motion if time flows from $t = T$ to $t = 0$. Given a sample $\boldsymbol{x}_T$ from the prior distribution, integrating Eq. (5) with initial condition $\boldsymbol{x}(T) = \boldsymbol{x}_T$ from $t = T$ to $t = 0$ results in a sample from $p(\boldsymbol{x})$. Moreover, to each reverse-SDE, there exists a corresponding deterministic reverse-ODE

$$d\boldsymbol{x} = \left[\boldsymbol{f}(\boldsymbol{x}, t) - (1/2) \cdot g(t)^2 \nabla_{\boldsymbol{x}} \log p_t(\boldsymbol{x})\right] dt \tag{6}$$

which also can be integrated from $t = T$ to $t = 0$ to produce samples from $p(\boldsymbol{x})$.

Diffusion models generate data by simulating the reverse of the diffusion process, i.e., by solving the reverse-S/ODE of the diffusion process. Initial works on diffusion models (Sohl-Dickstein et al., 2015; Ho et al., 2020) used computationally expensive ancestral sampling to solve the reverse differential equations. Later works discovered that using adaptive numerical integrators to solve the reverse-S/ODE could accelerate the sampling process. This led to great attention on developing better reverse-S/ODE integrators (Jolicoeur-Martineau et al., 2021; Song et al., 2021b; Lu et al., 2022; Karras et al., 2022; Zhang & Chen, 2022). Our work is orthogonal to such works as focus on finding good initialization points for integration via MCMC. Hence, a better integration technique directly translates to even better generative performance when plugged into Denoising MCMC.

**Variance exploding (VE) diffusion model.** A VE diffusion model considers the diffusion process

$$d\boldsymbol{x} = \sqrt{\frac{d[\sigma^2(t)]}{dt}}\, d\boldsymbol{w}. \tag{7}$$

where $\sigma(t)$ increases from $\sigma_{\min} = \sigma(0)$ to $\sigma_{\max} = \sigma(T)$. Distribution of $\boldsymbol{x}(t)$ evolves as

$$p_t(\boldsymbol{x}) = \int p_{\sigma(t)}(\boldsymbol{x} \mid \tilde{\boldsymbol{x}}) p(\tilde{\boldsymbol{x}})\, d\tilde{\boldsymbol{x}} \tag{8}$$

so if $\sigma_{\min}$ is sufficiently small, $p_0(\boldsymbol{x}) \approx p(\boldsymbol{x})$, and if $\sigma_{\max}$ is sufficiently large, so variance explodes, $p_T(\boldsymbol{x}) \approx \mathcal{N}(\boldsymbol{x} \mid \boldsymbol{0}, \sigma_{\max}^2 \boldsymbol{I})$. If we have a score model $\boldsymbol{s}_\theta(\boldsymbol{x}, \sigma)$ trained with Eq. (2), $\nabla_{\boldsymbol{x}} \log p_t(\boldsymbol{x}) \approx \boldsymbol{s}_\theta(\boldsymbol{x}, \sigma(t))$. It follows that with $\boldsymbol{x}_T \sim \mathcal{N}(\boldsymbol{x} \mid \boldsymbol{0}, \sigma_{\max}^2 \boldsymbol{I})$, we may integrate the reverse-S/ODE corresponding to Eq. (7) with $\boldsymbol{x}(T) = \boldsymbol{x}_T$ from $t = T$ to $t = 0$ using a score model to generate data. In the next section, we bridge MCMC and reverse-S/ODE integrators with VE diffusion to form a novel sampling framework that improves both MCMC and diffusion models.

## 3 Denoising Markov Chain Monte Carlo (DMCMC)

We now develop a general framework called Denoising MCMC (DMCMC) which combines MCMC with reverse-S/ODE integrators. We denote the data space as $\mathcal{X} \subseteq \mathbb{R}^d$ and the noise level space as $\mathcal{S} = [\sigma_{\min}, \sigma_{\max}]$. The construction of DMCMC is comprised of two steps. In the first step, we build MCMC to generate initialization points in the product space $\mathcal{X} \times \mathcal{S}$, i.e., $\mathcal{X}$ augmented by the smoothness parameter $\sigma$. Since $\sigma(t)$ is a monotone increasing function, this is equivalent to augmenting the data space with diffusion time $t$. In the second step, we incorporate denoising steps, where we denoise the generated initialization points via reverse-S/ODE integrators. Note that while the predictor-corrector (PC) method (Song et al., 2021b) also uses MCMC, PC uses MCMC to improve the denoising process. Thus, DMCMC and PC contribute to orthogonal aspects of diffusion sampling. More discussion on the differences between DMCMC and PC is in Appendix D.

### 3.1 Construction Step 1: MCMC on the Product Space $\mathcal{X} \times \mathcal{S}$

Suppose $p(\boldsymbol{x})$ is a high-dimensional multimodal distribution, supported on a low-dimensional manifold. If the modes are separated by wide low-density regions, MCMC can have difficulty moving between the modes. Indeed, convergence time for such distributions can grow exponential in dimension $d$ (Roberts & Rosenthal, 2001). Intuitively, for MCMC to move between disjoint modes, the Markov Chain would have to step off the data manifold. However, once MCMC leaves the data manifold, the density or the score vanishes. Then, most random directions produced by the proposal distribution do not point to the manifold. Thus, MCMC gets lost in the ambient space, whose volume grows exponentially in $d$.

Annealing via Gaussian smoothing, used in both ALD and VE diffusion, circumvents this problem. As $p(\boldsymbol{x})$ smoothed with perturbation kernel $p_\sigma(\boldsymbol{x} \mid \tilde{\boldsymbol{x}})$ of increasing $\sigma$, the modes of $p(\boldsymbol{x})$ grow wider and start to connect. Thus, MCMC can easily transition between modes. However, running MCMC in the manner of ALD is inefficient since we do not know how many iterations within each noise level is sufficient. To address this problem, we propose to augment $\mathcal{X}$ with the smoothness scale $\sigma$ and run MCMC in the product space $\mathcal{X} \times \mathcal{S}$ such that MCMC automatically controls the value of $\sigma$. Below, we formally describe MCMC on $\mathcal{X} \times \mathcal{S}$.

Let us define the $\sigma$-conditional distribution
$$\hat{p}(\boldsymbol{x} \mid \sigma) \coloneqq \int p_\sigma(\boldsymbol{x} \mid \tilde{\boldsymbol{x}}) p(\tilde{\boldsymbol{x}}) \, d\tilde{\boldsymbol{x}}. \tag{9}$$
We also define a prior $\hat{p}(\sigma)$ on $\mathcal{S}$. Then by the Bayes' Rule,
$$\hat{p}(\boldsymbol{x}, \sigma) = \hat{p}(\boldsymbol{x} \mid \sigma) \cdot \hat{p}(\sigma). \tag{10}$$
In Appendix E.2, we discuss the effect of varying the prior. MCMC with $\hat{p}(\boldsymbol{x}, \sigma)$ will produce samples $\{(\boldsymbol{x}_n, \sigma_n)\}$ in $\mathcal{X} \times \mathcal{S}$ such that
$$\sigma_n \sim \hat{p}(\sigma), \qquad \boldsymbol{x}_n \sim \hat{p}(\boldsymbol{x} \mid \sigma_n). \tag{11}$$
Hence, if $\sigma_n \gg \sigma_{\min}$, $\boldsymbol{x}_n$ will be a noisy sample, i.e., a sample corrupted with Gaussian noise of variance $\sigma_n^2$, and if $\sigma_n \approx \sigma_{\min}$, $\boldsymbol{x}_n$ will resemble a sample from $p(\boldsymbol{x})$.

Since our goal is to generate samples from $p(\boldsymbol{x})$, naïvely, we can keep samples $(\boldsymbol{x}_n, \sigma_n)$ with $\sigma_n \approx \sigma_{\min}$ and discard other samples. However, this could lead to a large waste of computation resources. In the next section, we incorporate reverse-S/ODE integrators to avert this problem.

### 3.2 Construction Step 2: Incorporating Denoising Steps

Let us recall that integrating the reverse-S/ODE for the VE diffusion SDE Eq. (7) from time $t = T$ to $t = 0$ sends samples from $p_T(\boldsymbol{x})$ to samples from $p_0(\boldsymbol{x}) \approx p(\boldsymbol{x})$. In general, integrating the reverse-SDE or ODE from time $t = t_2$ to $t = t_1$ for $t_1 < t_2$ sends samples from $p_{t_2}(\boldsymbol{x})$ to samples from $p_{t_1}(\boldsymbol{x})$ (Song et al., 2021b). We use this fact to denoise MCMC samples from $\hat{p}(\boldsymbol{x}, \sigma)$.

Suppose we are given a sample $(\boldsymbol{x}_n, \sigma_n) \sim \hat{p}(\boldsymbol{x}, \sigma)$. With $t_n \coloneqq \sigma^{-1}(\sigma_n)$, Eq. (11) tells us
$$\boldsymbol{x}_n \sim p_{t_n}(\boldsymbol{x}) \tag{12}$$
so integrating the reverse-S/ODE with initial condition $\boldsymbol{x}(t_n) = \boldsymbol{x}_n$ from $t = t_n$ to $t = 0$ produces a sample from $p_0(\boldsymbol{x}) \approx p(\boldsymbol{x})$. Here, we note that any reverse-S/ODE solver may be used to carry out the integration. In Appendix F.2, we show the necessity of the denoising step.

Let us briefly explain how DMCMC accelerates sampling. Given an MCMC chain $\{(\boldsymbol{x}_n, \sigma_n)\}$ in $\mathcal{X} \times \mathcal{S}$ and a prior $\hat{p}(\sigma)$ which places high mass near $\sigma_{\min}$, MCMC traverses high probability regions, so $\sigma_n \ll \sigma_{\max}$ for most $n$, i.e., $t_n \ll T$ for most $n$. This also means the sequence $\{\boldsymbol{x}_n\}$ generally stays close to the data manifold. So, the average length of integration intervals $(0, t_n)$ will tend to be much shorter than $T$. Thus, integrating the reverse-S/ODE over $(0, t_n)$ to denoise $\boldsymbol{x}_n$ is much faster than integrating the reverse-S/ODE over $(0, T)$, i.e., standard diffusion sampling. This idea is illustrated in Figure 1, and more detailed explanation is given in Appendix E.1.

## 4 Denoising Langevin Gibbs (DLG)

In Section 3, we described an abstract framework, DMCMC, for accelerating score-based sampling by combining MCMC and reverse-S/ODE integrators. We now develop a concrete instance of DMCMC. As the second construction step of DMCMC is simple, we only describe the first step.

Naïvely, we could extend denoising score matching Eq. (1) to estimate the score $\hat{s}_\theta(\boldsymbol{x}, \sigma) : \mathbb{R}^d \times \mathbb{R} \to \mathbb{R}^d \times \mathbb{R}$ of $\hat{p}(\boldsymbol{x}, \sigma)$ and apply Langevin dynamics in the first step of DMCMC. But, this would prevent us from using pre-trained score models, as we would have to solve (for some small $\nu > 0$)

$$\min_\theta \mathbb{E}_{\boldsymbol{\epsilon} \sim \mathcal{N}(\boldsymbol{0}_{d+1}, \nu^2 \boldsymbol{I}_{d+1})} \mathbb{E}_{\hat{p}(\boldsymbol{x}, \sigma)}[\|\hat{s}_\theta(\boldsymbol{x} - \epsilon_{1:d}, \sigma - \epsilon_{d+1}) - \boldsymbol{\epsilon}/\nu^2\|_2^2] \tag{13}$$

which differs from Eq. (2). Gibbs sampling provides a simple path around this problem. Let us recall that given a previous MCMC iterate $(\boldsymbol{x}_n, \sigma_n)$, Gibbs sampling proceeds by alternating between an $\boldsymbol{x}$ update step $\boldsymbol{x}_{n+1} \sim \hat{p}(\boldsymbol{x} \mid \sigma_n)$ and a $\sigma$ update step $\sigma_{n+1} \sim \hat{p}(\sigma \mid \boldsymbol{x}_{n+1})$. Below, we describe our score-based sampling algorithm, Denoising Langevin Gibbs (DLG) (pseudocode in Appendix C).

**Updating $\boldsymbol{x}$.** Suppose we are given an MCMC iterate $(\boldsymbol{x}_n, \sigma_n)$ and a score model $s_\theta(\boldsymbol{x}, \sigma)$ from Eq. (2). We generate $\boldsymbol{x}_{n+1}$ by a Langevin dynamics step on $\hat{p}(\boldsymbol{x} \mid \sigma_n)$. Specifically, by Eq. (9),

$$\nabla_{\boldsymbol{x}} \log \hat{p}(\boldsymbol{x} \mid \sigma_n) \approx s_\theta(\boldsymbol{x}, \sigma_n) \tag{14}$$

and so an Langevin dynamics update on $\boldsymbol{x}$, according to Eq. (3) is

$$\boldsymbol{x}_{n+1} = \boldsymbol{x}_n + (\eta/2) \cdot s_\theta(\boldsymbol{x}_n, \sigma_n) + \sqrt{\eta} \cdot \boldsymbol{\epsilon} \tag{15}$$

for $\boldsymbol{\epsilon} \sim \mathcal{N}(\boldsymbol{0}, \boldsymbol{I})$. Here, we call $\eta$ the step size.

**Updating $\sigma$.** We now have $\boldsymbol{x}_{n+1}$ and need to sample $\sigma_{n+1} \sim \hat{p}(\sigma \mid \boldsymbol{x}_{n+1})$. To this end, we first train a DNN noise level classifier $q_\phi(\sigma \mid \boldsymbol{x})$ to approximate $\hat{p}(\sigma \mid \boldsymbol{x})$ by solving

$$\max_\phi \mathbb{E}_{\hat{p}(\boldsymbol{x}, \sigma)}[\log q_\phi(\sigma \mid \boldsymbol{x})]. \tag{16}$$

Specifically, we discretize $[\sigma_{\min}, \sigma_{\max}]$ into $M$ levels $\tau_1 = \sigma_{\min} < \tau_2 < \cdots < \tau_M = \sigma_{\max}$. Given $\tau_m$ where $1 \leq m \leq M$, $m$ serves as the label and clean training data corrupted by Gaussian noise of variance $\tau_m^2$ serves as the classifier input. The classifier is trained to predict $m$ by minimizing the cross entropy loss. Having trained a noise level classifier, we sample $\sigma_{n+1}$ by drawing an index $m$ according to the classifier output probability for $\boldsymbol{x}_{n+1}$ and setting $\sigma_{n+1} = \tau_m$. In practice, using the index of largest probability worked fine. We denote this process as $\sigma_{n+1} \sim q_\phi(\sigma \mid \boldsymbol{x}_{n+1})$. In Appendix F.1 we verify whether DLG with the approximated conditional works as intended.

### 4.1 PRACTICAL CONSIDERATIONS

**Computation cost of $\sigma$ prediction.** We found that using shallow neural networks for the noise classifier $q_\phi$ was sufficient to accelerate sampling. Concretely, using a neural net with four convolution layers and one fully connected layer as the classifier, one evaluation of $q_\phi$ was around $100 \sim 1000$ times faster than one evaluation of the score model $\boldsymbol{s}_\theta$. So, when comparing sampling methods, we only count the number of score function evaluations (NFE). We also note that the training time $q_\phi$ was negligible compared to the training time of $s_\theta$. For instance, on CelebA-HQ-256, training $q_\phi$ with the aforementioned architecture for 100 epochs took around 15 minutes on an RTX 2080 Ti.

**Starting points for DLG.** Theoretically, an MCMC chain $\{(\boldsymbol{x}_n, \sigma_n)\}_{n=0}^\infty$ will converge to $\hat{p}(\boldsymbol{x}, \sigma)$ regardless of the starting point $(\boldsymbol{x}_0, \sigma_0)$. However, theory shows that setting starting points close to the stationary distribution, i.e., using "warm start", can significantly accelerate convergence of the Markov chain (Dalalyan, 2017; Dwivedi et al., 2019). Thus, we set $\boldsymbol{x}_0$ by generating a clean data sample with a reverse-S/ODE solver starting from prior noise, adding some Gaussian noise to the clean data sample, and running Gibbs sampling for a few iterations. Pseudocode is shown in Appendix C. The NFE involved in generating $\boldsymbol{x}_0$ is included in the final per-sample average NFE computation for DLG when comparing methods in Section 5. But, we note that this cost vanishes in the limit of infinite sample size.

**Reducing autocorrelation.** Autocorrelation in MCMC chains, i.e., correlation between consecutive samples in the MCMC chain, could reduce the sample diversity of MCMC. A typical technique to reduce autocorrelation is to use every $n_{skip}$-th samples of the MCMC chain for some $n_{skip} > 1$. For DMCMC, this means we denoise every $n_{skip}$-th sample. So, if we use $n_{den}$ NFE to denoise MCMC samples, the average NFE for generating a single sample is around $n_{skip} + n_{den}$.

**Choosing iterates to apply denoising.** The MCMC chain can be partitioned into blocks which consist of $n_{skip}$ consecutive samples. Using every $n_{skip}$-th sample of the MCMC chain corresponds to denoising the last iterate of each block. Instead, to further shorten the length of integration, within each block, we apply denoising to the sample of minimum noise scale $\sigma$.

**Choice of prior $\hat{p}(\sigma)$.** We use $\hat{p}(\sigma) \propto 1/\sigma$ to drive the MCMC chain towards small values of $\sigma$.

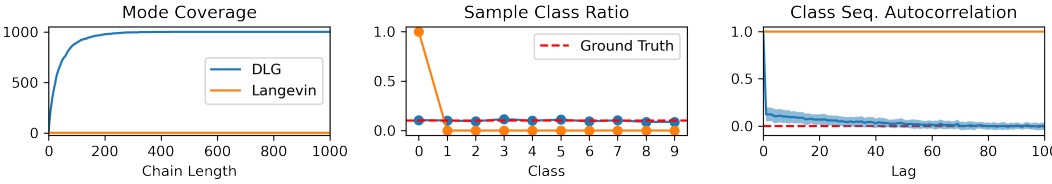

Figure 2: Ablation study of $\sigma$ update step in DLG.

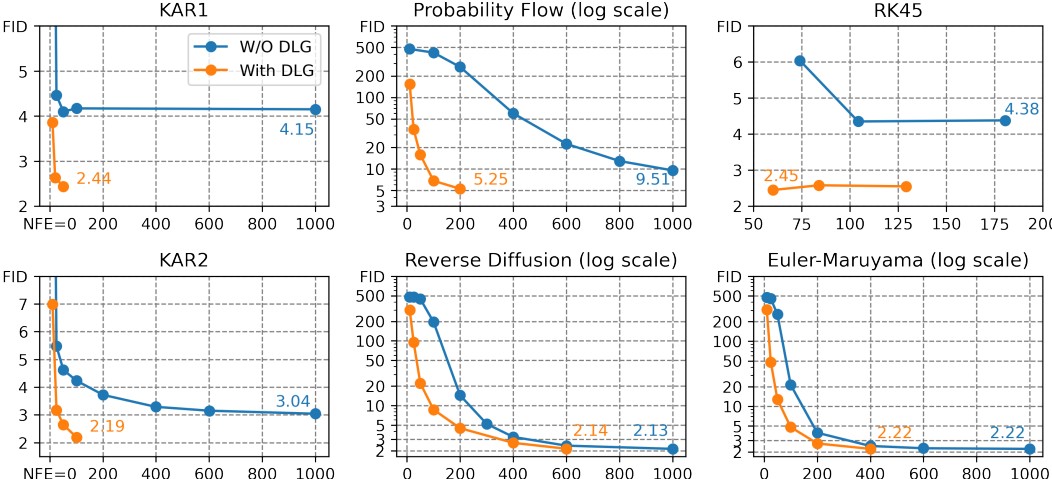

Figure 3: Sampling acceleration of DLG on CIFAR10. FID of notable points are written in the corresponding color. **Top row**: deterministic integrators. **Bottom row**: stochastic integrators.

# 5 EXPERIMENTS

## 5.1 MIXING OF DMCMC CHAINS

Here we provide experimental proof that DMCMC is capable of visiting diverse modes as a consequence of running MCMC in the product space $\mathcal{X} \times \mathcal{S}$. To this end, we compare DLG with and without $\sigma$ updates. DLG without $\sigma$ updates is just Langevin dynamics at fixed $\sigma$, so we run fifty Langevin dynamics chains and fifty DLG chains on a mixture of Gaussians (MoG) with $1k$ modes at CIFAR10 images. All chains are initialized at a single mode. For each method, we compute the mode coverage of the samples, the class distribution of the samples, and the autocorrelation of sample image class sequence. Since the noise conditional score function can be calculated analytically for MoGs, this setting decouples sampler performance from score model approximation error.

Figure 2 shows the results. In the left panel, we observe that Langevin dynamics is unable to escape the initial mode. Increasing the step size $\eta$ of Langevin dynamics caused the chain to diverge. On the other hand, DLG successfully captures all modes of the distribution. DLG samples cover all $1k$ modes at chain length $432$. Middle panel provides evidence that DLG samples correctly reflect the statistics of the data distribution. Finally, the right panel indicates that the DLG chain moves freely between classes, i.e., distant modes. These observations validate our claim that DLG mixes well.

## 5.2 ACCELERATING IMAGE GENERATION WITH SCORE NETWORKS

We compare six integrators with and without DLG on CIFAR10 and CelebA-HQ-256 image generation. The deterministic integrators are: the deterministic integrator of Karras et al. (2022) (KAR1), the probability flow integrator of Song et al. (2021b), and the RK45 solver. The stochastic integrators are: the stochastic integrator of Karras et al. (2022) (KAR2), the reverse diffusion integrator of Song et al. (2021b), and the Euler-Maruyama method. We use the Fréchet Inception Distance (FID) (Heusel et al., 2017) to measure sample quality. For CIFAR10, we generate $50k$ samples, and for CelebA-HQ-256, we generate $10k$ samples. We use pre-trained score models of Song et al. (2021b).

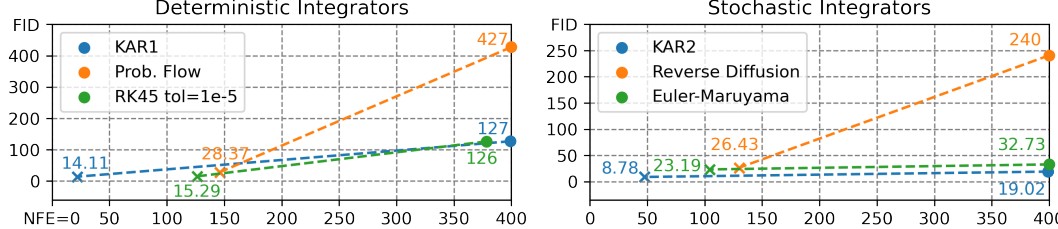

Figure 4: Sampling acceleration of DLG on CelebA-HQ-256. A dot indicates an integrator without DLG, and a cross of the same color indicates corresponding integrator combined with DLG. Dotted lines indicate performance improvement due to DLG.

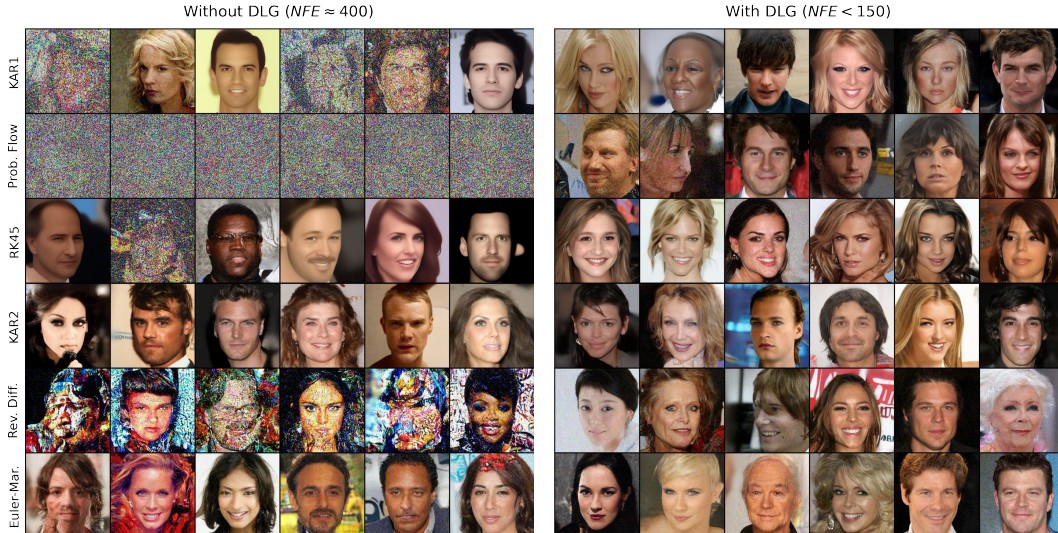

Figure 5: Non-cherry-picked samples on CelebA-HQ-256 using the settings for Fig. 4. Each row shows samples for an integrator without (left col.) and with (right col.) DLG.

**CIFAR10.** In Figure 3, we make two important observations. First, DLG successfully accelerates all six integrators by a non-trivial margin. In particular, if an integrator without DLG already performs well, the integrator combined with DLG outperforms other integrators combined with DLG. For instance, compare the results for KAR1 with those of other deterministic integrators. Second, DLG improves the performance lower bound for some deterministic integrators. While the performance of KAR1 and RK45 saturate at around 4 FID, KAR1 and RK45 with DLG achieve around 2.4 FID.

| Method | NFE 10 | NFE 20 | NFE 50 |
|---|---|---|---|
| DPM-Solver-2 (VP) | 5.28 (+2 NFE) | 3.02 (+4 NFE) | 2.69 (−2 NFE) |
| DPM-Solver-3 (VP) | 6.03 (+2 NFE) | 2.75 (+4 NFE) | 2.65 (−2 NFE) |
| DEIS (VP) | 4.17 (+0 NFE) | 2.86 (+0 NFE) | 2.57 (+0 NFE) |
| DEIS (VE) | 20.89 (+0 NFE) | 16.59 (+0 NFE) | 16.31 (+0 NFE) |
| KAR1 (VP) | 9.70 (+1 NFE) | 3.23 (+5 NFE) | 2.97 (+1 NFE) |
| KAR1 (VE) | 14.12 (+1 NFE) | 4.46 (+5 NFE) | 4.1 (+1 NFE) |
| DLG+KAR1 (VE) | **3.86** (+0.1 NFE) | **2.63** (+0.1 NFE) | **2.45** (−0.9 NFE) |

Table 1: Comparison of fast samplers on CI-FAR10 FID. Number in parenthesis indicates extra or less NFE used. Best numbers are bolded.

**CelebA-HQ-256.** Figure 4 shows the results on CelebA-HQ-256. We observe that DLG improves computational efficiency and sample quality simultaneously. Indeed, in Figure 5, we observe remarkable improvements in sample quality despite using fewer NFE. This demonstrates the scalability of DLG to generating high-resolution images. We also note that we did not perform an exhaustive search of DLG hyper-parameters for CelebA-HQ-256, so fine-tuning could yield better results.

**Achieving SOTA.** DLG combined with KAR1 sets a new SOTA record for CIFAR10 in the limited number of NFE setting: 3.86 FID with 10.11 NFE and 2.63 FID with 20.11 NFE which beats the results of Zhang & Chen (2022), 4.17 FID with 10 NFE and 2.86 FID with 20 NFE. DLG combined with KAR2 sets a new record on CelebA-HQ-256 among score-based models: 6.99 FID with 158.96 NFE which beats the current best result of Kim et al. (2022), 7.16 FID with 4000 NFE.

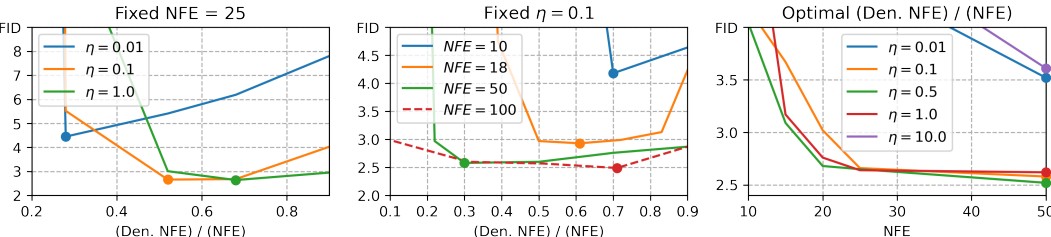

Figure 6: Ablation study of DLG with KAR1. Dots indicate the points of lowest FID.

### 5.3 HYPER-PARAMETER ABLATION STUDY

Given an integrator, DLG is determined by total NFE per sample $n$, NFE spent on denoising samples $n_{den}$, and Langevin dynamics step size $\eta$. We fix the integrator to be KAR1 and observe the effect of each component on CIFAR10 image generation. We observed similar trends with other samplers.

**$\eta$ vs. $n_{den}/n$.** In the left panel of Figure 6, we fix NFE and vary $\eta$ and $n_{den}/n$. $n_{den}$ governs individual sample quality, and $n_{skip} = n - n_{den}$ governs sample diversity. Thus, we observe optimal FID is achieved when $n_{den}/n$ has intermediate values, not extreme values near 0 or 1. Also, lower $n_{den}/n$ is needed to attain optimality for lower $\eta$. This is because lower $\eta$ means the MCMC chain travels closer to the image manifold at the cost of slower mixing.

**NFE vs. $n_{den}/n$.** In the middle panel of Figure 6, we fix $\eta$ and vary NFE and $n_{den}/n$. We observe two trends. First, in the small NFE regime, where $10 \leq$ NFE $\leq 50$, it is beneficial to decrease $n_{den}/n$ as NFE increases. Second, in the large NFE regime, where NFE $> 50$, it is beneficial to increase $n_{den}/n$ as NFE increases. This is because if $n_{skip}$ is sufficiently large, MCMC chain starts producing essentially independent samples, so increasing $n_{skip}$ further provides no gain.

Also, as we increase NFE, the set of $n_{den}/n$ which provides near-optimal performance becomes larger. Moreover, we see that most of the time, optimal FID is achieved when $n_{den}/n > 0.5$, i.e., when $n_{den} > n_{skip}$. So, a reasonable strategy for choosing $n_{den}$ given $\eta$ and NFE budget $n$ is to find smallest $n_{skip}$ which produces visually distinct samples, and then allocate $n_{den} = n - n_{skip}$.

**$\eta$ vs. NFE.** In the right panel of Figure 6, we choose optimal (in terms of FID) $n_{den}/n$ for each combination of $\eta$ and NFE. We see choosing overly small or large $\eta$ leads to performance degradation. If $\eta$ is within a certain range, we obtain similarly good performance. In the case of CIFAR10, we found it reasonable to set $\eta \in [0.05, 1.0]$.

To choose $\eta$ for data of general dimension $d$, we define a value $\kappa := \eta/\sqrt{d}$ called displacement per dimension. If we see Eq. (14), Gaussian noise of zero mean and variance $\eta$ is added to the sample at each update step. Gaussian annulus theorem tells us that a high-dimensional Gaussian noise with zero mean and variance $\eta$ has Euclidean norm approximately $\eta\sqrt{d}$. So, the average displacement of the sample per dimension by the random noise is around $\eta/\sqrt{d}$. Since $\kappa$ is a dimension-independent value, given $\kappa$ and $d$, we can set $\eta = \sqrt{d}\kappa$. On CIFAR10, we have $\eta \in [0.05, 1.0]$, which translates to $\kappa \in [0.0009, 0.018]$. This means, on CelebA-HQ-256, we can choose $\eta \in [0.4, 8.0]$. If the sampler was inefficient, we chose a smaller $\kappa$ to trade-off diversity for sample quality.

## 6 CONCLUSION

In this work, we proposed DMCMC which combines MCMC with reverse-S/ODE integrators. This has led to improvements for both MCMC and diffusion models. For MCMC, DMCMC allows Markov chains to visit disjoint modes. For diffusion models, DMCMC accelerates sampling by reducing the average integration interval length of reverse-S/ODE. We developed a practical instance of DMCMC called DLG, and demonstrated the practicality and scalability of DLG through various experiments. In particular, DLG achieved state-of-the-art results on CIFAR10 and CelebA-HQ-256 among score-based models. Overall, our work opens up an orthogonal approach to accelerating score-based sampling. We leave exploration of other kinds of MCMC or diffusion process such as VP diffusion (see Appendix E.3) in DMCMC as future work.

# 7 REPRODUCIBILITY STATEMENT

We provide an anonymous Google drive link to a zip file containing code and noise classifier checkpoints for our main experiments. https://drive.google.com/file/d/1C5RO4UB6x8eatIHDxGiGLV_oLsEKcyOI/view?usp=share_link We have also added a pseudocode of DMCMC in Appendix C, and hyper-parameters are described in Appendix A.

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

## A DETAILED EXPERIMENT SETTINGS

**Device.** We use an RTX 2080 Ti or two Quadro RTX 6000 depending on the required VRAM.

**Codes.** For probability flow, RK45, reverse diffusion, and Euler-Maruyama integrators, we modify the code provided by Song et al. (2021b) in the GitHub repository `https://github.com/yang-song/score_sde_pytorch`. For KAR1 and KAR2, since Karras et al. (2022) did not release their implementation of the samplers, we used our implementation based on their paper. For evaluation, we use the FID implementation provided in the GitHub repository `https://github.com/mseitzer/pytorch-fid`.

**Datasets.** We use the CIFAR10 dataset (Krizhevsky, 2009) and the CelebA-HQ-256 dataset (Karras et al., 2018).

**Data processing.** All data are normalized into the range $[0, 1]$. Following Song et al. (2021b), for all methods, a denoising step using Tweedie's denoising formula is applied at the end of the sampling process.

**Noise predictor network** $q_\phi$**.** The noise predictor network $q_\phi$ has four convolution layers followed by a fully connected layer. The convolution layers have channels 32, 64, 128, 256, and $(\sigma_{\min}, \sigma_{\max})$ is discretized into $1k$ points $\sigma_{\min}(\sigma_{\max}/\sigma_{\min})^t$ for $t$ spaced evenly on $[0, 1]$. On CIFAR10, $q_\phi$ is trained for 200 epochs, and on CelebA-HQ-256, $q_\phi$ is trained for 100 epochs. We use the Adam optimizer (Kingma & Ba, 2015) with learning rate $0.001$.

**Mixture of Gaussians.** MoG has $1k$ modes at randomly sampled CIFAR10 training set images. For Langevin dynamics, we use step size $\eta = 0.0001$. Using a larger step size caused the Langevin dynamics chain to diverge. For DLG, we use step size $\eta = 1.0$, $n_{skip} = 1$, $n_{den} = 20$, and the reverse diffusion integrator. For each method, all chains were initialized at a single mode to test mixing capabilities in the worst-case scenario.

**CIFAR10 image generation.** We use a pre-trained NCSN++ (cont.) score model provided by Song et al. (2021b). For the baseline methods, we use the recommended settings. For DLG, the chain was initialized by generating samples with the deterministic integrator of Karras et al. (2022) using 37 NFE, adding Gaussian noise of variance 0.25, and running 20 iterations of Langevin-Gibbs. Table 2 lists the hyper-parameters for DLG and the corresponding FID used to produce Figure 3.

| KAR1 | | | | Probability Flow | | | | RK45 | | | |
|---|---|---|---|---|---|---|---|---|---|---|---|
| $n_{den}$ | $n_{skip}$ | $\eta$ | FID | $n_{den}$ | $n_{skip}$ | $\eta$ | FID | $n_{den}$ | $n_{skip}$ | $\eta$ | FID |
| 9 | 1 | 0.5 | 3.86 | 9 | 1 | 0.01 | 153.48 | 59.98 | 10 | 0.5 | 2.45 |
| 17 | 3 | 0.5 | 2.63 | 24 | 1 | 0.01 | 35.58 | 83.74 | 10 | 0.5 | 2.58 |
| 27 | 24 | 0.5 | 2.44 | 49 | 1 | 0.01 | 15.75 | 128.99 | 10 | 0.5 | 2.55 |
| | | | | 90 | 10 | 0.01 | 6.8 | | | | |
| | | | | 190 | 10 | 0.05 | 5.25 | | | | |

| KAR2 | | | | Reverse Diffusion | | | | Euler-Maruyama | | | |
|---|---|---|---|---|---|---|---|---|---|---|---|
| $n_{den}$ | $n_{skip}$ | $\eta$ | FID | $n_{den}$ | $n_{skip}$ | $\eta$ | FID | $n_{den}$ | $n_{skip}$ | $\eta$ | FID |
| 9 | 1 | 0.5 | 6.99 | 9 | 1 | 0.01 | 300.18 | 9 | 1 | 0.01 | 306.98 |
| 23 | 2 | 0.5 | 3.17 | 24 | 1 | 0.01 | 94.98 | 24 | 1 | 0.01 | 47.83 |
| 31 | 20 | 0.5 | 2.64 | 49 | 1 | 0.01 | 22.1 | 49 | 1 | 0.02 | 12.73 |
| 51 | 50 | 0.5 | 2.19 | 90 | 10 | 0.01 | 8.61 | 90 | 10 | 0.05 | 4.79 |
| | | | | 190 | 10 | 0.1 | 4.45 | 190 | 10 | 0.1 | 2.69 |
| | | | | 370 | 30 | 1.0 | 2.64 | 370 | 30 | 1.0 | 2.22 |
| | | | | 570 | 30 | 1.0 | 2.14 | | | | |

Table 2: DLG hyper-parameters and FID for integrators in Figure 3.

**CelebA-HQ-256 image generation.** We use a pre-trained NCSN++ (cont.) score model provided by Song et al. (2021b). For the baseline methods, we use the recommended settings. For DLG, the chain was initialized by generating samples with the stochastic integrator of Karras et al. (2022) using 37 NFE, adding Gaussian noise of variance 0.25, and running 70 iterations of Langevin-Gibbs. Table 3 lists the hyper-parameters for DLG and the corresponding FID used to produce Figure 4.

| KAR1 | | | | Probability Flow | | | | RK45 | | | |
|---|---|---|---|---|---|---|---|---|---|---|---|
| $n_{den}$ | $n_{skip}$ | $\eta$ | FID | $n_{den}$ | $n_{skip}$ | $\eta$ | FID | $n_{den}$ | $n_{skip}$ | $\eta$ | FID |
| 17 | 4 | 0.8 | 14.11 | 140 | 5 | 0.25 | 28.37 | 125.95 | 4 | 0.8 | 15.29 |

| KAR2 | | | | Reverse Diffusion | | | | Euler-Maruyama | | | |
|---|---|---|---|---|---|---|---|---|---|---|---|
| $n_{den}$ | $n_{skip}$ | $\eta$ | FID | $n_{den}$ | $n_{skip}$ | $\eta$ | FID | $n_{den}$ | $n_{skip}$ | $\eta$ | FID |
| 37 | 10 | 0.8 | 8.78 | 100 | 25 | 0.1 | 26.43 | 100 | 4 | 0.8 | 23.19 |

Table 3: DLG hyper-parameters and FID for deterministic samplers in Figure 4.

**Achieving SOTA.** On CIFAR10, we use KAR1 settings of Table 2. On CelebA-HQ-256, we use $n_{den} = 131$, $n_{skip} = 27$, $\eta = 4.0$, which achieves 6.99 FID.

## B  ADDITIONAL SAMPLES

### B.1  DLG CHAIN VISUALIZATION

Figures 7 and 8 each show a DLG chain on CIFAR10 and CelebA-HQ-256, respectively. The chain progresses from left to right, from right end to left end of row below. On CIFAR10, we can see that the chain visits diverse classes. On CelebA-HQ-256, we can see that the chain transitions between diverse attributes such as gender, hair color, skin color, glasses, facial expression, posture, etc.

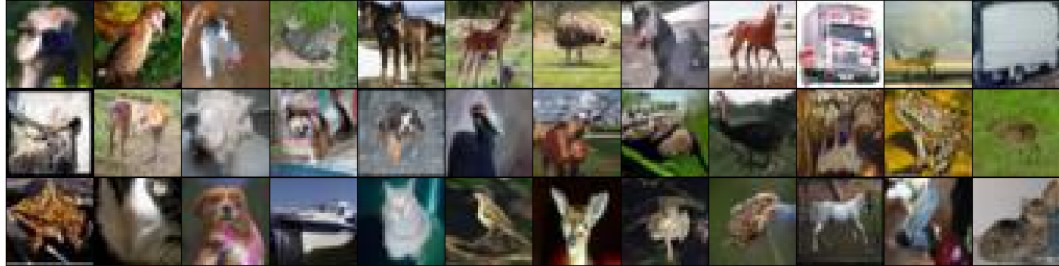

Figure 7: Visualization of a DLG chain on CIFAR10.

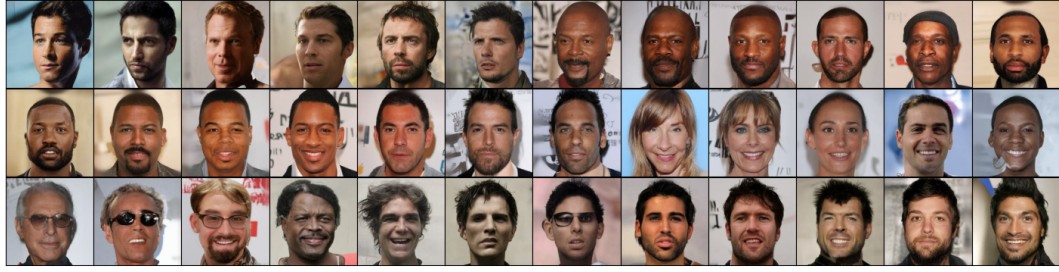

Figure 8: Visualization of a DLG chain on CelebA-HQ-256.

### B.2  ADDITIONAL UNCONDITIONAL SAMPLES

In Figures 9 and 10, we show additional samples for CIFAR10 without and with DLG. In Figures 11 and 12, we show additional samples for CelebA-HQ-256 without and with DLG.

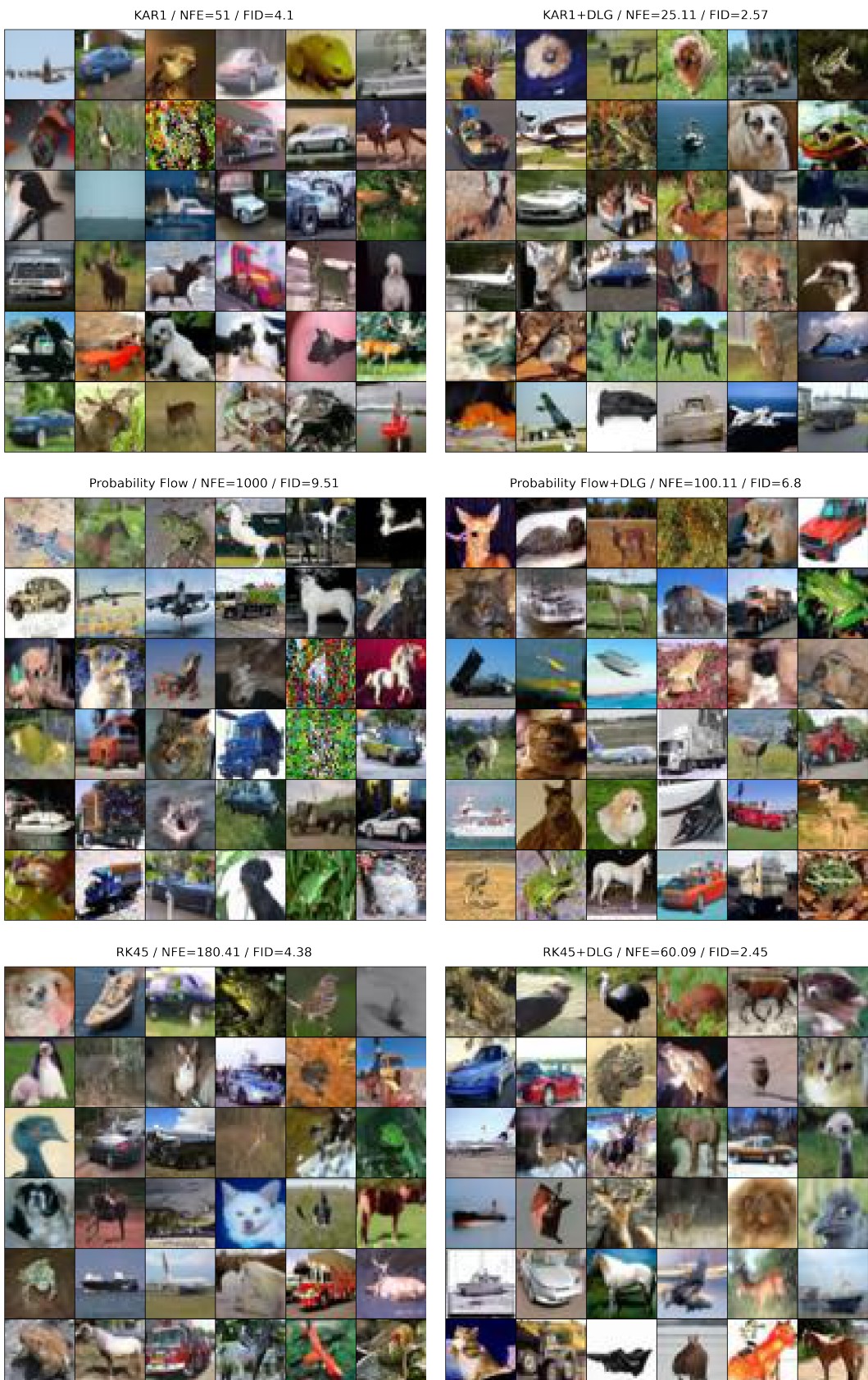

Figure 9: Additional non-cherry-picked samples for deterministic integrators on CIFAR10 without (left col.) and with (right col.) DLG.

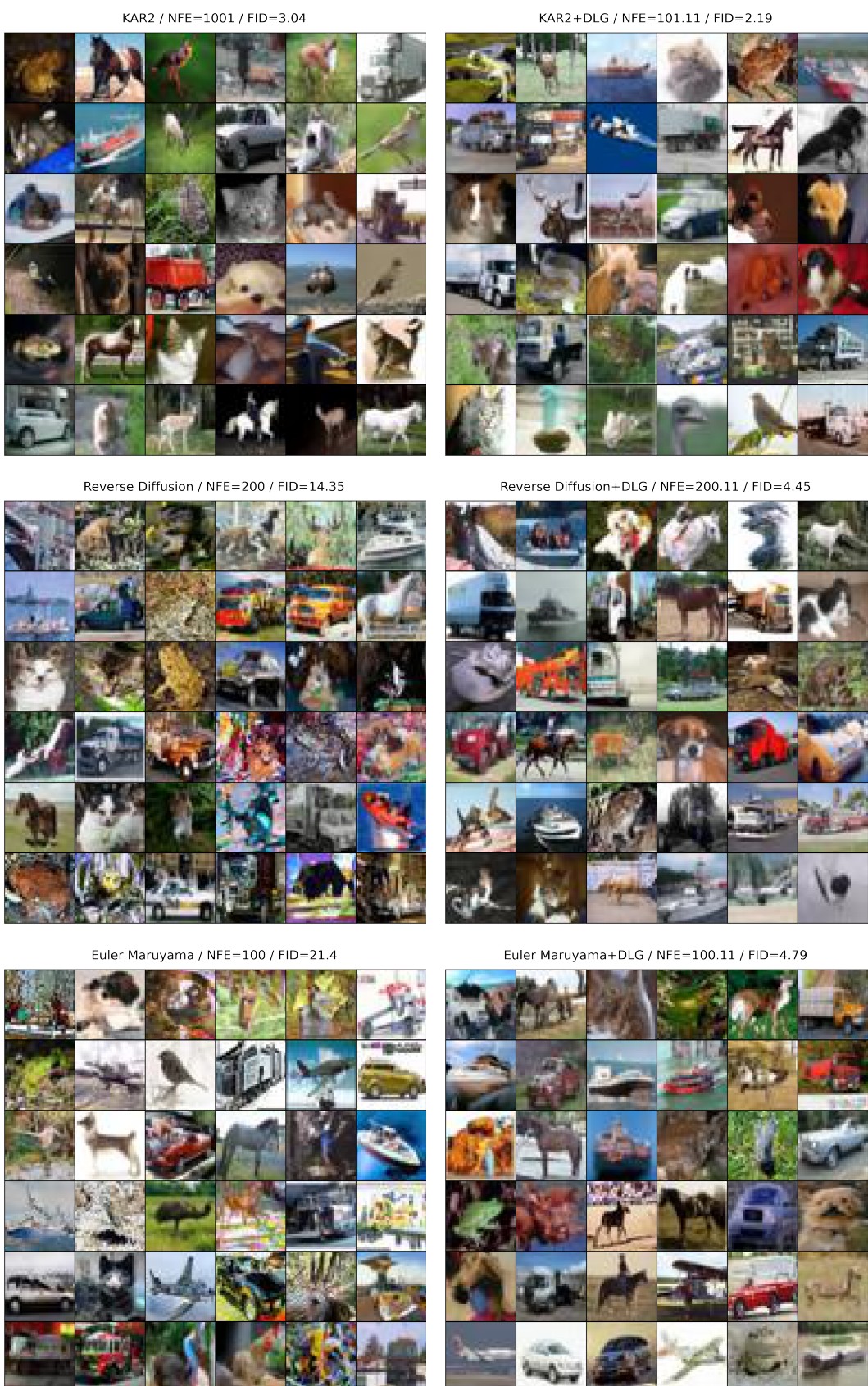

Figure 10: Additional non-cherry-picked samples for stochastic integrators on CIFAR10 without (left col.) and with (right col.) DLG.

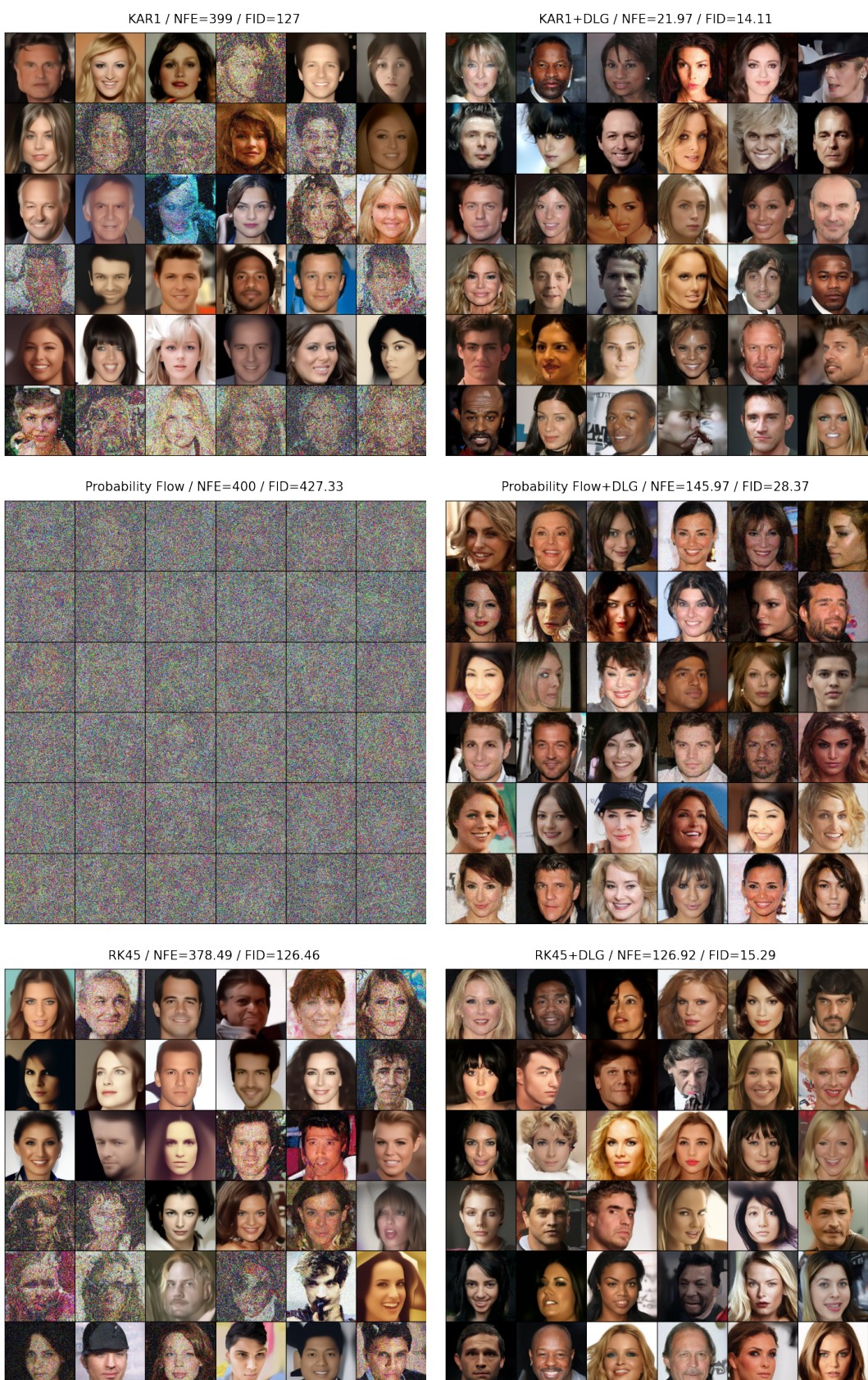

Figure 11: Additional non-cherry-picked samples for deterministic integrators on CelebA-HQ-256 without (left col.) and with (right col.) DLG.

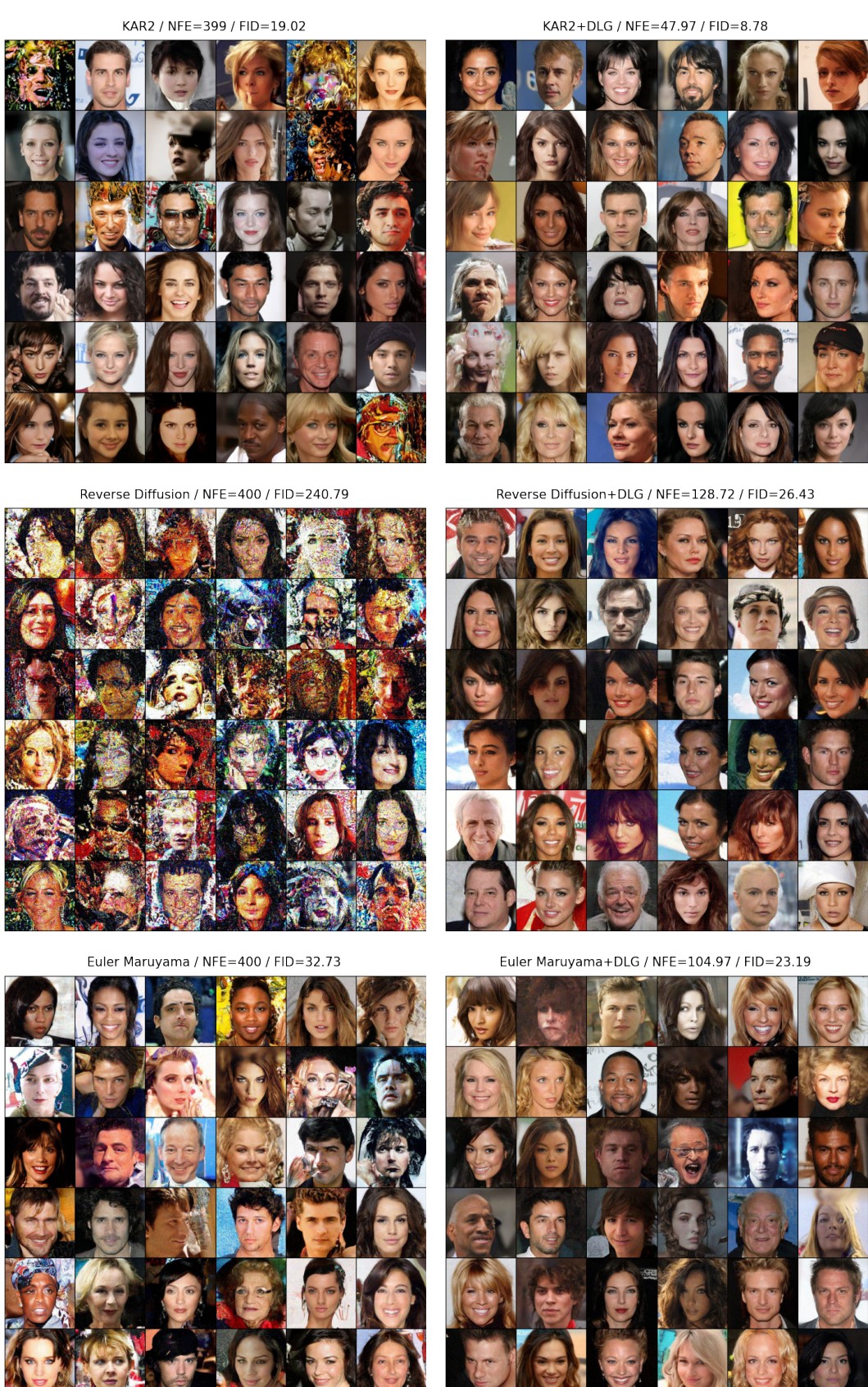

Figure 12: Additional non-cherry-picked samples for stochastic integrators on CelebA-HQ-256 without (left col.) and with (right col.) DLG.

## C    PSEUDOCODES

Let us denote a reverse-S/ODE integrator as $\pi(s_\theta, \boldsymbol{x}, \sigma, n)$. Given a point $\boldsymbol{x}$ at noise level $\sigma$, score function $s_\theta$, and NFE budget $n$, $\pi$ returns the result of integrating reverse-S/ODE from $\sigma$ to $\sigma_{\min}$ starting from $\boldsymbol{x}$. That is, $\pi$ returns a sample from $p_0(\boldsymbol{x})$.

---

**Algorithm 1** MCMC Starting Point Generation

---

1: **Input:** Integration NFE budget $n_1$, number of Langevin-Gibbs steps $n_2$, Langevin step size $\eta$
2: Sample $\tilde{\boldsymbol{x}}_0 \sim \mathcal{N}(\boldsymbol{0}, \sigma_{\max}^2 \boldsymbol{I})$
3: $\tilde{\boldsymbol{x}}_0 \leftarrow \pi(s_\theta, \tilde{\boldsymbol{x}}_0, \sigma_{\max}, n_1)$
4: $\tilde{\boldsymbol{x}}_0 \leftarrow \tilde{\boldsymbol{x}}_0 + 0.5 \cdot \boldsymbol{\epsilon}$ where $\boldsymbol{\epsilon} \sim \mathcal{N}(\boldsymbol{0}, \boldsymbol{I})$
5: $\sigma_0 \leftarrow 0.5$
6: **for** $t = 1, 2, \ldots n_2$ **do**
7:      $\tilde{\boldsymbol{x}}_0 \leftarrow \tilde{\boldsymbol{x}}_0 + 0.5 \cdot \eta \cdot s_\theta(\tilde{\boldsymbol{x}}_0, \sigma_0) + \sqrt{\eta} \cdot \boldsymbol{\epsilon}$ where $\boldsymbol{\epsilon} \sim \mathcal{N}(\boldsymbol{0}, \boldsymbol{I})$
8:      $\sigma_0 \sim q_\phi(\sigma \mid \boldsymbol{x}_0)$
9: **end for**
10: **Return** $(\tilde{\boldsymbol{x}}_0, \sigma_0)$

---

---

**Algorithm 2** Denoising Langevin Gibbs

---

1: **Input:** $n_{den}$, $n_{skip}$, MCMC starting point $(\tilde{\boldsymbol{x}}_0, \sigma_0)$, Langevin step size $\eta$, number of samples needed $N$
2: $(\tilde{\boldsymbol{x}}, \sigma) \leftarrow (\tilde{\boldsymbol{x}}_0, \sigma_0)$
3: **for** $k = 1, 2, \ldots N$ **do**
4:      Initialize minimum noise level tracking variables $(\hat{\boldsymbol{x}}, \hat{\sigma}) \leftarrow (\tilde{\boldsymbol{x}}, \sigma)$
5:      **for** $t = 1, 2, \ldots, n_{skip}$ **do**
6:          $\tilde{\boldsymbol{x}} \leftarrow \tilde{\boldsymbol{x}} + 0.5 \cdot \eta \cdot s_\theta(\tilde{\boldsymbol{x}}, \sigma) + \sqrt{\eta} \cdot \boldsymbol{\epsilon}$ where $\boldsymbol{\epsilon} \sim \mathcal{N}(\boldsymbol{0}, \boldsymbol{I})$
7:          $\sigma \sim q_\phi(\sigma \mid \tilde{\boldsymbol{x}})$
8:          **if** $\sigma < \hat{\sigma}$ **then**
9:              $(\hat{\boldsymbol{x}}, \hat{\sigma}) \leftarrow (\tilde{\boldsymbol{x}}, \sigma)$
10:          **end if**
11:      **end for**
12:      $\boldsymbol{x}_k \leftarrow \pi(s_\theta, \hat{\boldsymbol{x}}, \hat{\sigma}, n_{den})$
13: **end for**
14: **Return** $\{\boldsymbol{x}_k\}_{k=1}^N$

---

## D    MORE RELATED WORKS

### D.1    COMPARISON OF DMCMC WITH THE PREDICTOR-CORRECTOR SAMPLER

As both DMCMC and the predictor-corrector (PC) algorithm (Song et al., 2021b) rely on MCMC, one may ask whether DMCMC is a straightforward extension of PC. However, we assure the reader that DMCMC is not a trivial generalization of PC.

On a high level, the main message of our paper is that we can significantly improve diffusion sampling by choosing better initial points for the denoising process. This claim was verified through extensive experiments. On the other hand, the focus of the corrector step of PC is on improving the denoising process by refining the predictor / integrator steps. So, DMCMC and PC contribute to orthogonal aspects of diffusion sampling. Indeed, the improvements offered by MCMC corrector step are rather marginal (see Table 1 in (Song et al., 2021b)) compared to the improvements offered by DMCMC (see Figure 3 in our paper). On a low level, we distinguish DMCMC from PC on three levels: theoretical, implementation-wise, and empirical.

Theoretically, MCMC in DMCMC and MCMC in PC reduce truncation error (error arising from numerically integrating reverse-S/ODE) in entirely distinct ways. We observe that diffusion model sampling can be broken down into two parts: (part 1) generating an initialization point, and (part 2) integrating the reverse-S/ODE starting from the initialization point to generate clean data. MCMC in PC aims to improve part 2 whereas MCMC in DMCMC aims to improve part 1.

In PC, MCMC is used as a corrector. That is, MCMC corrects the distributions of intermediate points during integration of the reverse-S/ODE (part 2). That is why PC sampling proceeds by alternating between taking a predictor step and running Langevin dynamics at a fixed noise level.

In DMCMC, MCMC generates initialization points for reverse-S/ODE (part 1) that lie near the image manifold. This is possible because MCMC runs in the augmented space $\mathcal{X} \times \mathcal{S}$ by adaptively updating the noise level. This leads to reduced truncation error, or equivalently, acceleration, as it is easier for reverse-S/ODE integrators to generate clean data from points near the data manifold than from prior noise distribution samples (e.g., Gaussian noise). For a more detailed explanation of the acceleration mechanism of DMCMC, we refer the reader to Appendix E.1.

DMCMC does not resemble PC even from the perspective of implementation. MCMC in PC runs during reverse-S/ODE integration, and MCMC in DMCMC runs before reverse-S/ODE integration:

- PC : (prior noise) $\rightarrow$ (integ. step) $\rightarrow$ (MCMC) $\rightarrow$ (integ. step) $\rightarrow$ (MCMC) $\rightarrow$ ... $\rightarrow$ (data)
- DMCMC : (MCMC to generate points near image manifold) $\rightarrow$ (integration) $\rightarrow$ (data)

Moreover, as already mentioned, MCMC in PC runs in $\mathcal{X}$ at a fixed $\sigma$ whereas MCMC in DMCMC runs in $\mathcal{X} \times \mathcal{S}$ by adaptively updating $\sigma$.

Finally, DMCMC can also be used to accelerate PC algorithms as well, as shown in Figure 13. This means MCMC in DMCMC and PC play orthogonal roles in generating clean data. If DMCMC were a straightforward extension of PC, we would not see such acceleration.

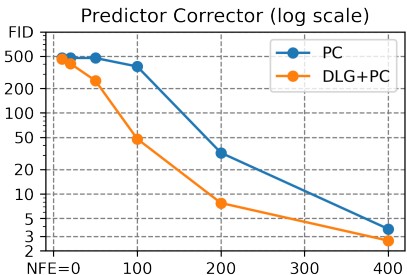

Figure 13: PC acceleration with DLG on CIFAR10.

## D.2 ON THE NOISE PREDICTOR NETWORK IN DMCMC

Nichol & Dhariwal (2021) and Roman et al. (2021) have also used noise predictor networks to improve diffusion sampling. Although both works employed a neural net to predict the noise level / diffusion time for a given data input, we emphasize that the works do not overlap with the contributions of DMCMC. The main contribution of our paper is not the noise prediction network itself. The novelty of DMCMC arises from how we use the noise prediction network.

On a high level, the main message of our paper is that we can significantly improve diffusion sampling by choosing better initial points for denoising. A noise predictor network was used in the process of finding initial points. On the other hand, the focus of the mentioned works is on improving the denoising process with a noise predictor network. So, DMCMC and the mentioned works contribute to orthogonal aspects of diffusion sampling.

Let us elaborate. We first note that diffusion model sampling can be broken down into two parts: (part 1) generating an initialization point, and (part 2) integrating the reverse-S/ODE to generate data from the initialization point.

Nichol & Dhariwal (2021) and Roman et al. (2021) use the noise prediction network to accelerate integration of the reverse-S/ODE (improve part 2). This is achieved by learning the covariance of the reverse distribution (Nichol & Dhariwal, 2021) or by adjusting the noise schedule with a neural net (Roman et al., 2021).

In DMCMC, MCMC generates initialization points for reverse-S/ODE that lie near the image manifold (improve part 1). This is possible because MCMC runs in $\mathcal{X} \times \mathcal{S}$ by adaptively updating the

noise level with a noise prediction network. This leads to acceleration, as it is easier for reverse-S/ODE integrators to generate clean data from points near the data manifold than from prior noise distribution samples (e.g., Gaussian noise). Hence, our paper proposes an acceleration approach entirely different from those of and Nichol & Dhariwal (2021) and Roman et al. (2021).

In fact, DMCMC does not resemble Nichol & Dhariwal (2021) and Roman et al. (2021) even from the perspective of implementation. Noise predictors in Nichol & Dhariwal (2021) and Roman et al. (2021) are used during reverse-S/ODE integration, and the noise predictor in DMCMC is used before reverse-S/ODE integration as a sub-step of MCMC.

## E MORE DISCUSSIONS

### E.1 MORE EXPLANATION ON HOW DMCMC ACCELERATES SAMPLING

The discussion at the end of Section 3.2 describes why DMCMC combined with a reverse-S/ODE integrator is faster than using a reverse-S/ODE integrator alone. Here, we provide a more detailed explanation of the acceleration phenomenon. To begin, we note that DMCMC consists of two steps that are executed sequentially: (a) MCMC on $\hat{p}(\boldsymbol{x}, \sigma)$, and (b) denoising MCMC samples by reverse-S/ODE. Since DMCMC without (a) is just standard diffusion, we see the acceleration behavior comes from using $\hat{p}(\boldsymbol{x}, \sigma)$ samples as initial points for reverse-S/ODE. Thus, we need to show how using $\hat{p}(\boldsymbol{x}, \sigma)$ samples as initial points for reverse-S/ODE accelerates image generation.

This proceeds in two steps. We first explain how MCMC produces samples $(\boldsymbol{x}_n, \sigma_n)$ from $\hat{p}(\boldsymbol{x}, \sigma)$ such that $\sigma_n$ is significantly smaller than $\sigma_{\max}$. Then, we explain how integrating over $(\sigma_{\min}, \sigma_n)$ is faster than integrating over $(\sigma_{\min}, \sigma_{\max})$ as in standard diffusion.

To begin, we first explain how MCMC produces samples $(\boldsymbol{x}_n, \sigma_n)$ from $\hat{p}(\boldsymbol{x}, \sigma)$ such that $\sigma_n$ is significantly smaller than $\sigma_{\max}$. We observe that $\int p_\sigma(\boldsymbol{x}|\tilde{\boldsymbol{x}})p(\tilde{\boldsymbol{x}})d\tilde{\boldsymbol{x}}$ becomes flatter and wider as $\sigma$ increases. This is because $\int p_\sigma(\boldsymbol{x}|\tilde{\boldsymbol{x}})p(\tilde{\boldsymbol{x}})d\tilde{\boldsymbol{x}}$ means we are applying Gaussian smoothing to $p(\boldsymbol{x})$ with a Gaussian kernel of variance $\sigma^2$. For instance, if $p(\boldsymbol{x})$ is a normal distribution with variance $\gamma^2$, $\int p_\sigma(\boldsymbol{x}|\tilde{\boldsymbol{x}})p(\tilde{\boldsymbol{x}})d\tilde{\boldsymbol{x}}$ is a normal distribution with variance $\sigma^2 + \gamma^2$. It follows that high density values of $\hat{p}(\boldsymbol{x}, \sigma)$ occur when $\boldsymbol{x}$ is near the data manifold and $\sigma$ is small. Since MCMC traverses high-probability regions, we can expect $\{\boldsymbol{x}_n\}$ will be close to the data manifold and $\{\sigma_n\}$ will be smaller than $\sigma_{\max}$. Indeed, in Appendix F.1, we see that actual $\sigma_n$ values are significantly smaller than $\sigma_{\max} = 50$ in CIFAR10.

Standard diffusion needs to integrate the reverse-S/ODE over the large interval $(\sigma_{\min}, \sigma_{\max})$ to produce clean images. On the other hand, in DMCMC, MCMC produces samples $(\boldsymbol{x}_n, \sigma_n)$ from $\hat{p}(\boldsymbol{x}, \sigma)$ such that $\sigma_n$ is significantly smaller than $\sigma_{\max}$. So, in the DMCMC framework, to generate clean images, we only need to integrate the reverse-S/ODE over the small interval $(\sigma_{\min}, \sigma_n)$. This means the cost of integrating over $(\sigma_n, \sigma_{\max})$ vanishes for DMCMC, leading to accelerated image generation. (To be precise, the cost of integrating over $(\sigma_n, \sigma_{\max})$ is replaced by the cost of running MCMC to sample from $\hat{p}(\boldsymbol{x}, \sigma)$, but we observe in Section 5.1 that DLG mixes rapidly, so this cost is negligible.)

More rigorously, given the same computation budget and the same integration method, integrating over $(\sigma_{\min}, \sigma_n)$ has less truncation error than integrating over $(\sigma_{\min}, \sigma_{\max})$. Computation budget roughly corresponds to the number of discretization points of the integration interval we can use to approximate the reverse-S/ODE. Thus, a shorter interval of integration means we can use smaller step size $h$ during integration, which implies smaller error. For instance, Euler's method has $O(h^2)$ local error. A more rigorous exposition is given in Chapter 7 of Stoer & Bulisch (2002).

This justifies how DMCMC can generate better samples than standard diffusion under a fixed computation budget. In other words, DMCMC can use less computation budget than standard diffusion to achieve similar sample quality as standard diffusion, i.e., DMCMC can accelerate sampling.

### E.2 TRADE-OFF BETWEEN THE CONVERGENCE SPEED OF MCMC AND SHARPNESS OF $\hat{p}(\sigma)$

Let us assume that $-\log \hat{p}(\boldsymbol{x} \mid \sigma)$ is strongly convex and has $L$ Lipschitz continuous gradients. We also assume $-\log \hat{p}(\sigma)$ is strongly convex and has $M$ Lipschitz continuous gradients. We use such assumptions, because in the setting where either $-\log \hat{p}(\boldsymbol{x} \mid \sigma)$ or $-\log \hat{p}(\sigma)$ is nonconvex, it is

difficult to say anything theoretically meaningful about convergence of MCMC on the joint $\hat{p}(\boldsymbol{x}, \sigma)$. We also assume the MCMC of choice is Langevin dynamics for ease of analysis.

The sharpness of the prior $\hat{p}(\sigma)$ can then be characterized by $M$. Intuitively, if $\hat{p}(\sigma)$ is more peaked around zero, $-\log \hat{p}(\sigma)$ will have a gradient which changes more rapidly, and so $M$ will be larger. We then note that since $-\log \hat{p}(\boldsymbol{x}, \sigma) = -\log \hat{p}(\boldsymbol{x} \mid \sigma) - \log \hat{p}(\sigma)$, $-\log \hat{p}(\boldsymbol{x}, \sigma)$ is also strongly convex and $-\log \hat{p}(\boldsymbol{x}, \sigma)$ has $L + M$ Lipschitz continuous gradients.

We now resort to Theorem 3 in (Cheng & Bartlett, 2018), which shows that the convergence time of Langevin dynamics on $-\log \hat{p}(\boldsymbol{x}, \sigma)$ in terms of KL divergence grows in the order of $O((L + M)^2)$ (ignoring log terms). So, we indeed see there is a trade-off between the mixing time of MCMC and the choice of the prior. Using a prior that is sharper around zero will increase $M$ and thus increase convergence time quadratically. We speculate that a similar analysis will hold for Langevin Gibbs as well, but a rigorous analysis of Langevin Gibbs is worthy of a paper of its own.

However, in practice, we do note that Langevin Gibbs with the prior used in our paper converges quite fast, as shown in Section 5.1. In the rightmost panel of Figure 2, we observe that the autocorrelation of image labels vanishes after only a few iterations. If the sampler mixed poorly in the $\boldsymbol{x}$ space, image labels would have high autocorrelation. For visualization of DLG chains, we refer the readers to Appendix B.1. So, we can say Langevin Gibbs reliably produces samples from $\hat{p}(\boldsymbol{x}, \sigma)$ even with a small number of steps.

### E.3 VE Diffusion vs. VP Diffusion in DMCMC

A natural question is whether we can join MCMC with reverse-S/ODE using the VP diffusion framework as well. It is true that the VP setting could be more stable than the VE setting, and that is why some recent solvers work in the VP setting. For instance, DPM-Solver (Lu et al., 2022) only provides experiment results in the VP setting, despite the fact that DPM-Solver can be applied to the VE setting as well. In the case of DEIS (Zhang & Chen, 2022), there is a large performance gap in the VE and VP settings. Specifically, the performance of DEIS deteriorates significantly in the VE setting.

We used VE diffusion because, from the perspective of generating better initialization points via MCMC, VE setting was better than VP setting. We believe the benefits outweigh the downsides, since, as shown in Table 1, DLG beats all recent fast solvers regardless of whether that fast solver works in the VP setting or the VE setting. We conjecture that stability does not have a large influence on DMCMC because the reverse-S/ODE initialization points generated by MCMC are close to the image manifold, such that the variance of initialization points is small compared to the variance of prior noise. This is possibly why DMCMC improves the performance lower bound for some deterministic integrators (Section 5.2).

We also give a detailed explanation of why DMCMC is more compatible with VE. Concretely, we observed two differences between VE diffusion and VP diffusion that made MCMC with VP diffusion difficult.

We first establish some notations. For $\alpha \in [0, 1)$, we define the perturbation kernel $p_\alpha(\boldsymbol{x} \mid \tilde{\boldsymbol{x}}) :=$ $\mathcal{N}(\boldsymbol{x} \mid \sqrt{\alpha}\tilde{\boldsymbol{x}}, (1 - \alpha)\boldsymbol{I})$ and the $\alpha$-conditional distribution $\hat{p}(\boldsymbol{x} \mid \alpha) := \int p_\alpha(\boldsymbol{x} \mid \tilde{\boldsymbol{x}})p(\tilde{\boldsymbol{x}}) \, d\tilde{\boldsymbol{x}}$. If $\alpha = 0$, we get the prior noise distribution, which is the standard normal distribution. If $\alpha = 1$, we recover the data distribution. VP diffusion proceeds by decreasing $\alpha$ from 1 to 0 (Song et al., 2021b). We can generate clean data from Gaussian noise by using a numerical integrator to solve the reverse-S/ODE for VP diffusion from $\alpha = 0$ to $\alpha = 1$. With a prior $\hat{p}(\alpha)$ on $\mathcal{A} := [0, 1)$, we then have a joint distribution

$$\hat{p}(\boldsymbol{x}, \alpha) = \hat{p}(\boldsymbol{x} \mid \alpha) \cdot \hat{p}(\alpha) \tag{17}$$

on the product space $\mathcal{X} \times \mathcal{A}$. Then, analogous to DMCMC with VE diffusion, we may use MCMC to produces samples $\{(\boldsymbol{x}_n, \alpha_n)\}$ from $\hat{p}(\boldsymbol{x}, \alpha)$ with the help of an $\alpha$-classifier $q_\phi$ and then integrate the reverse-S/ODE from $\alpha_n$ to 1 to generate clean data. However, we observed two differences between VE diffusion and VP diffusion that made MCMC with VP diffusion difficult.

In VE diffusion, the support[1] of $\int p_\sigma(\boldsymbol{x} \mid \tilde{\boldsymbol{x}})p(\tilde{\boldsymbol{x}})d\tilde{\boldsymbol{x}}$ grows wider as $\sigma$ increases. Thus, wherever the current MCMC $\boldsymbol{x}$ iterate is, there always is some noise level $\sigma$ such that $\boldsymbol{x}$ is in the support of $\int p_\sigma(\boldsymbol{x} \mid \tilde{\boldsymbol{x}})p(\tilde{\boldsymbol{x}})d\tilde{\boldsymbol{x}}$ so the score function provides a meaningful (i.e., non-zero) direction. The noise classification network $q_\phi$ predicts this optimal noise level $\sigma$ for $\boldsymbol{x}$, as shown in Appendix F.1.

On the other hand, in the VP diffusion framework, the prior noise distribution (standard normal distribution) has finite support. To be precise, the standard normal distribution has non-zero density everywhere, but numerically, density becomes zero at points sufficiently far from the origin. As the data distribution also has finite support, all intermediate distributions $\hat{p}(\boldsymbol{x} \mid \alpha)$ of VP diffusion have finite support as well. Since MCMC takes random steps, it is possible for an MCMC iterate to land at a point where no intermediate distribution $\hat{p}(\boldsymbol{x} \mid \alpha)$ of VP diffusion provides meaningful density or gradient. Even worse, the mean of the corruption process of VP diffusion shifts in the process of changing data samples to zero mean standard normal samples. So, high-density paths between prior noise samples and data samples can become very narrow in high-dimensional scenarios.

Due to these pathologies of VP diffusion, what we observed happening in practice was that Langevin Gibbs would repeat the following behavior: try to approach the image manifold in the $\boldsymbol{x}$ update step so $\alpha$ increases, step off the high-density path leading to the data manifold in the $\boldsymbol{x}$ update step, and get absorbed into the prior noise distribution so $\alpha$ decays to 0. Indeed, in Figure 14 we observe $\alpha$ values oscillating around zero. This implies $\boldsymbol{x}_n$ are essentially prior noise samples, so using $\{(\boldsymbol{x}_n, \alpha_n)\}$ as initial points for the reverse-S/ODE has no practical benefit.

This type of problem with narrow, high-density regions often plagues MCMC. For a toy example, see Figure 3 of Cobb et al. (2019). We postulate that a better MCMC sampling scheme could enable us to run DMCMC under the VP diffusion framework. For instance, one could use MCMC with rejection steps that would reject points that step off high-density regions, or incorporate Riemannian manifold structure into the sampling scheme. However, we believe this is a topic for future work.

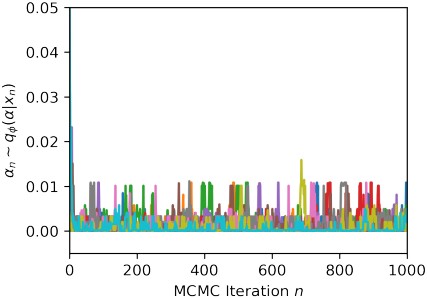

Figure 14: $\alpha_n$ trajectories for DLG with VP diffusion in the MoG setting.

# F  MORE EXPERIMENTS

## F.1  EVOLUTION OF $\sigma$ DURING LANGEVIN GIBBS ON $\mathcal{X} \times \mathcal{S}$

In Figure 15, we have visualized Langevin Gibbs trajectories of $\sigma$ in the MoG setting with $1k$ modes at CIFAR10 images (this setting decouples sampler behavior from score function approximation error) and the score network setting. We indeed observe that $\sigma$ moves up and down, allowing $\boldsymbol{x}$ to travel between disjoint modes of the distribution. Moreover, in Figure 2 of Section 5.1, $\boldsymbol{x}$ samples cover all $1k$ modes of the MoG. This experimentally proves $\boldsymbol{x}$ sequence of DLG is exploring the entire image distribution.

In the MoG setting, let us define $\delta_n$ as the distance of $\boldsymbol{x}_n$ to the closest mode in the MoG. In Figure 16, we see $\sigma_n \sim q_\phi(\sigma \mid \boldsymbol{x}_n)$ is almost identical to $\delta_n/\sqrt{d}$, where $d$ is the dimension of $\boldsymbol{x}_n$. This

---

[1] When we mention the support of a distribution, we mean the set of points where the numerical value (value represented on a computer) of the density is nonzero. This is because, even when a distribution theoretically has nonzero density everywhere, it is possible that the numerical value of the density vanishes outside some bounded region.

is reasonable, as the Gaussian annulus theorem tells us samples from a high-dimensional Gaussian distribution of mean $\mu$ and variance $\sigma^2$ come from a shell of radius $\sigma\sqrt{d}$ centered at $\mu$.

Specifically, due to the Gaussian annulus theorem, the samples of $\int p_\sigma(\boldsymbol{x} \mid \tilde{\boldsymbol{x}})p(\tilde{\boldsymbol{x}})d\tilde{\boldsymbol{x}}$ are most likely to come from a shell of radius $\sigma\sqrt{d}$ centered around the image manifold. (See Figure 2 (a) in Chung et al. (2022)) Then, given certain $\boldsymbol{x}$ which is distance $\delta$ from the image manifold, we can intuitively argue that optimal $\sigma$ for $\boldsymbol{x}$, i.e., $\sigma$ of highest likelihood under $p(\sigma \mid \boldsymbol{x})$, can be determined by equating $\delta = \sigma\sqrt{d}$ such that $\sigma = \delta/\sqrt{d}$. In the MoG case, this $\delta$ is approximated by the distance of $\boldsymbol{x}$ to the closest mode in the MoG. Since the $\sigma$ values predicted by the noise classifier agree with approximated $\delta/\sqrt{d}$, we can see that the noise classifier is a good approximation of $p(\sigma \mid \boldsymbol{x})$.

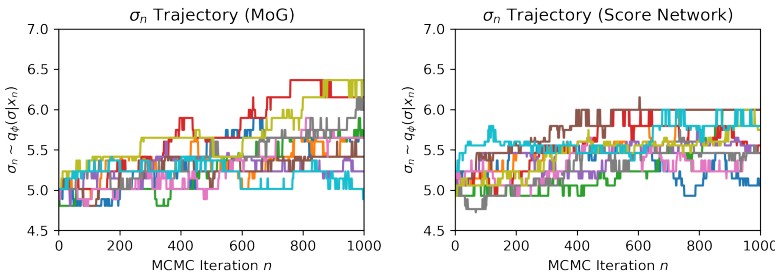

Figure 15: Visualization of $\sigma_n$ trajectories in MoG and score network settings.

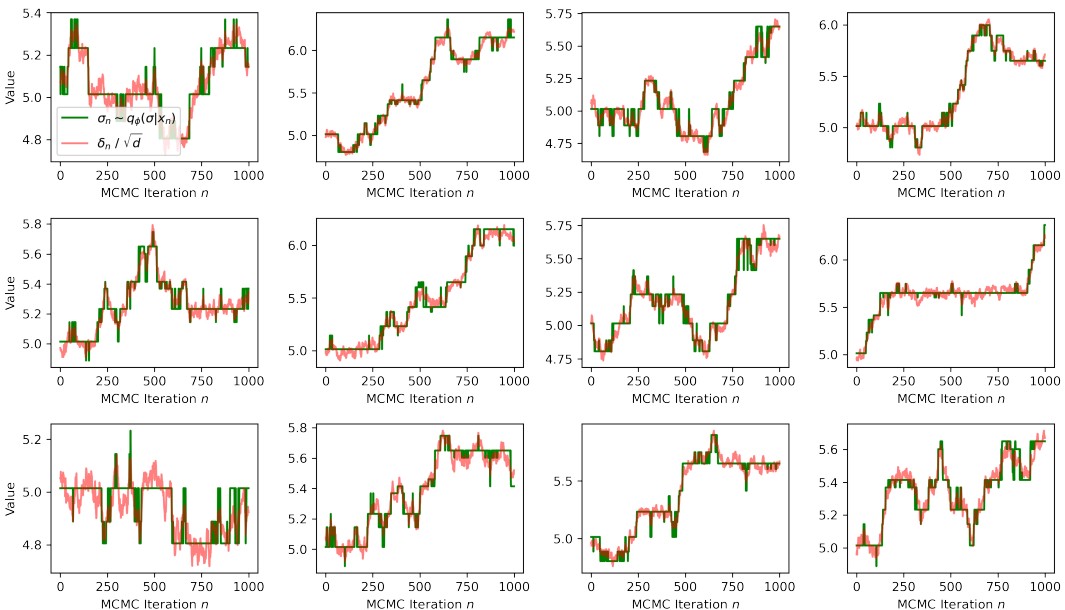

Figure 16: $q_\phi$ prediction and (approximate) distance to image manifold in the MoG setting.

## F.2 DENOISING STEP ABLATION

One may ask whether the denoising step in DMCMC is a necessary component to generate clean samples. To answer this question, we compare DLG with and without the denoising step. We use the reverse diffusion integrator as the reverse-SDE solver in the denoising step. Figure 17 shows DLG without the denoising step completely fails to generate valid clean image samples, regardless of how large $n_{skip}$ we use.

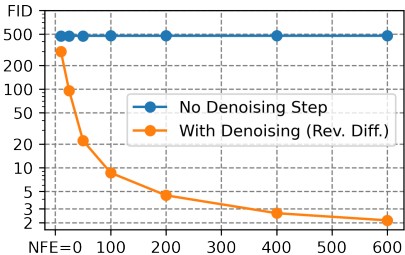

Figure 17: Ablation of the denoising step in DMCMC.

