# OpenReview forum: "Denoising MCMC for Accelerating Diffusion-Based Generative Models"
_ICLR.cc/2023/Conference — Submitted to ICLR 2023_

### Official Review · Reviewer_DAST · 2022-10-18

**Confidence:** 4
**Correctness:** 3
**Technical Novelty And Significance:** 3
**Empirical Novelty And Significance:** 3
**Recommendation:** 6

**Clarity, Quality, Novelty And Reproducibility:**

The paper is written clearly and well organized. Combining MCMC with reverse integrators is new for accelerating diffusion models.

**Strength And Weaknesses:**

Stength:

1. The paper is well written and organized.
2. The idea of using MCMC on the augment space for accelerating diffusion models is interesting and new.


Weaknesses:

1. The intialization of $x_0$ for DLG is not clearly stated. The author claimed that it needs to be set close to the image manifold. However, they also say $x_0$ is generated by sampling from a noise distribution, adding Gaussian noise of variance $0.25$, and runing Gibbs sampling a few iterations. I am confused with how $x_0$ is generated in practice. If it is generated by adding Gaussian noise to samples from real image data, the same trick can be applied to general diffusion models for acceleration which is not included in the ablation study.

2. I think the comparison to LD in section 5.1 is unfair. It would be better to compare with ALD with appropriate annealing schedules.

3. The biaseness towards high-density regions of $\mathcal{X}$ is controlled by the prior $\hat{p}(\sigma)$. Without appropriate choice of $\hat{p}(\sigma)$, the claim in section 3.2 needs not to hold.

4. Usually, MCMC takes some iterations for convergence. In practice, we want to use prior $\hat{p}(\sigma)$ the encourages small noise. There seems to be a trade-off between the mixing time of MCMC and the choice of $\hat{p}(\sigma)$ which is not discussed in this paper.

**Summary Of The Paper:**

This paper proposed a new way of acceleration diffusion based generative models that combines MCMC on a augmented space with reverse-S/ODE integrators. More specifically, a Gibbs sampler procedure is introduced to traverse the augmented space based on a pre-trained classifier. Numerical experiments demonstate the effectiveness of the proposed method.

**Summary Of The Review:**

The idea of using MCMC to accelerate reverse integration of diffusion model is interesting and experiments show positive evidence.

---

> ### Author Response · Authors · 2022-11-12
> **Response to Reviewer DAST (Part 2/2)**
>
> **Comment 4. Usually, MCMC takes some iterations for convergence. In practice, we want to use prior $\hat{p}(\sigma)$ that encourages small noise. There seems to be a trade-off between the mixing time of MCMC and the choice of $\hat{p}(\sigma)$ which is not discussed in this paper.**
>
> We thank the Reviewer for the important observation. The following discussion has been added as Appendix E.2 in our revised paper.
>
> Let us assume that $-\log \hat{p}(\mathbf{x} \mid \sigma)$ is strongly convex and has $L$ Lipschitz continuous gradients. We also assume $-\log \hat{p}(\sigma)$ is strongly convex and has $M$ Lipschitz continuous gradients. We use such assumptions, because in the setting where either $-\log \hat{p}(\mathbf{x} \mid \sigma)$ or $-\log \hat{p}(\sigma)$ is nonconvex, it is difficult to say anything theoretically meaningful about convergence of MCMC on the joint $\hat{p}(\mathbf{x},\sigma)$. We also assume the MCMC of choice is Langevin dynamics for ease of analysis.
>
> The sharpness of the prior $\hat{p}(\sigma)$ can then be characterized by $M$. Intuitively, if $\hat{p}(\sigma)$ is more peaked around zero, $-\log \hat{p}(\sigma)$ will have a gradient which changes more rapidly, and so $M$ will be larger. We then note that since $-\log \hat{p}(\mathbf{x},\sigma) = -\log \hat{p}(\mathbf{x} \mid \sigma) - \log \hat{p}(\sigma)$, $-\log \hat{p}(\mathbf{x},\sigma)$ is also strongly convex and $-\log \hat{p}(\mathbf{x},\sigma)$ has $L+M$ Lipschitz continuous gradients.
>
> We now resort to Theorem 3 in [1], which shows that the convergence time of Langevin dynamics on $-\log \hat{p}(\mathbf{x},\sigma)$ in terms of KL divergence grows in the order of $O((L+M)^2)$ (ignoring log terms). So, we indeed see there is a trade-off between the mixing time of MCMC and the choice of the prior. Using a prior that is sharper around zero will increase $M$ and thus increase convergence time quadratically. We speculate that a similar analysis will hold for Langevin Gibbs as well, but a rigorous analysis of Langevin Gibbs is worthy of a paper of its own.
>
> However, in practice, we do note that Langevin Gibbs with the prior used in our paper converges quite fast, as shown in Section 5.1. In the rightmost panel of Figure 2 of Section 5.1, we observe that the autocorrelation of image labels vanishes after only a few iterations. If the sampler mixed poorly in the $\mathbf{x}$ space, image labels would have high autocorrelation. For visualization of DLG chains, we refer the readers to Appendix B.1. So, we can say Langevin Gibbs reliably produces samples from $\hat{p}(\mathbf{x},\sigma)$ even with a small number of steps.
>
> **References**
>
> [1] Convergence of Langevin MCMC in KL-divergence, Machine Learning Research, 2018.

---

> ### Author Response · Authors · 2022-11-12
> **Response to Reviewer DAST (Part 1/2)**
>
> **Comment 1. The intialization of $x_0$ for DLG is not clearly stated. The author claimed that it needs to be set close to the image manifold. However, they also say $x_0$ is generated by sampling from a noise distribution, adding Gaussian noise of variance 0.25, and running Gibbs sampling a few iterations. I am confused with how $x_0$ is generated in practice. If it is generated by adding Gaussian noise to samples from real image data, the same trick can be applied to general diffusion models for acceleration which is not included in the ablation study.**
>
> To avoid confusion between initialization points for DLG and initialization points for reverse-S/ODE, we shall refer to initialization points for DLG as “starting points”.
>
> We clarify that we do not assume access to the training data or real image data. So, the trick of generating clean data by adding Gaussian noise to real image data mentioned by the Reviewer is not feasible in our setting. In our experiments, as now clarified in Section 4.1, MCMC starting point $\mathbf{x}_0$ is generated by first generating synthetic data samples by standard diffusion, adding Gaussian noise, and running Gibbs sampling for a few iterations. We have revised Section 4.1 of our paper to clarify how we generate $\mathbf{x}_0$. We also have added a pseudocode for $\mathbf{x}_0$ generation process and DLG in Appendix C. Moreover, we have also added a reproducibility statement (Section 7), which contains an anonymized link to our main experiment codes.
>
> **Comment 2. I think the comparison to LD in section 5.1 is unfair. It would be better to compare with ALD with appropriate annealing schedules.**
>
> We inform the Reviewer that the purpose of Section 5.1 is to show DLG is capable of visiting disjoint modes, not to show DLG is superior to standard Langevin dynamics. In other words, Section 5.1 is an ablation study rather than a comparison with Langevin dynamics as a baseline method. Specifically, in Section 5.1, we aim to show running DLG in $\mathcal{X} \times \mathcal{S}$ allows DLG to visit disjoint modes. To achieve this, we need to show that DLG without the $\sigma$ axis is incapable of visiting disjoint modes. Ablating the $\sigma$ axis corresponds to removing the $\sigma$ update step in DLG. DLG without the $\sigma$ update step, i.e., with a fixed $\sigma$, is standard Langevin dynamics at a fixed noise level (see Section 4), and that is why we compare DLG with Langevin dynamics in Section 5.1. We have clarified the above points in Section 5.1.
>
> Since the Reviewer asked, we also compared DLG with ALD (as a baseline method). ALD with NCSNv2 (Song et al.) achieves FID 10.87 on CIFAR-10 with 1000 NFE whereas DLG achieves 3.86 FID with around 10 NFE. This clearly shows that DLG is superior to ALD in terms of sampling performance and efficiency with an appropriate choice of integrator.
>
> **Comment 3. The biaseness towards high-density regions of $\hat{p}(\sigma)$ is controlled by the prior. Without appropriate choice of $\hat{p}(\sigma)$, the claim in section 3.2 needs not to hold.**
>
> We do not consider this point a weakness, since we already show with extensive experiments that $\hat{p}(\sigma) \propto 1/\sigma$ provides significant acceleration. However, we acknowledge that without an appropriate choice of the prior, the claim in Section 3.2 may not hold. We have modified our paper accordingly to reflect this point.

---

> ### Author Response · Authors · 2022-11-16
> **Do our revisions and responses answer your concerns and questions?**
>
> Dear Reviewer DAST,
>
> We thank the Reviewer for the constructive comments. As the end of the paper revision period is approaching, we would like to ask whether our paper revisions and responses have addressed your concerns and questions adequately. If not, we would be happy to discuss and update our paper further.
>
> Regards,
>
> the authors.

---

> > ### Author Response · Authors · 2022-12-05
> > **We are looking forward to your feedback**
> >
> > Dear Reviewer DAST,
> >
> > As there is only one week left until the end of the author-reviewer discussion stage, we would like to ask again whether our paper revisions and responses have addressed your concerns and questions adequately. If not, we would be happy to discuss further.
> >
> > Regards,
> >
> > the authors.

---

### Official Review · Reviewer_vzaX · 2022-10-24

**Confidence:** 3
**Correctness:** 3
**Technical Novelty And Significance:** 3
**Empirical Novelty And Significance:** 3
**Recommendation:** 6

**Clarity, Quality, Novelty And Reproducibility:**

Clarity: The paper is mostly clear, though I think it would be better to include a pseudocode in the paper.
Quality: The technical quality is good and the results are decent.
Novelty: Sampling from the joint space differs from previous approaches of building faster solvers.
Originality: To the best of my knowledge, the proposed approach is novel. Though I could possibly miss some relevant works.

**Strength And Weaknesses:**

Strength:
- Sampling from diffusion models is an important problem;
- Sampling in the joint space is somewhat novel, and I am somewhat surprised that the authors are able to make that work.
- The proposed sampler is efficient.

Weaknesses:
- How does the proposed method work with more recent fast solvers, such as DPM-Solver and DEIS? I wonder whether the proposed approach can still help if the solver itself is fast enough.
- The noise predictor network is a trained distribution rather than the true conditional p(sigma | x). With such an approximate conditional, MCMC can no longer guarantee to converge to the true stationary distribution (so rigorously it is not a Gibbs sampling algorithm). I wonder to what extent does this approximation affect the convergence / sample quality.
- It is not that intuitive to me how sampling in the joint space can be much faster than annealed MCMC. Doesn't the sampler still require many steps in the x space to mix? It would be better if the authors can better explain the mechanism behind the acceleration behavior. I understand that sampling from annealed distributions is better than directly sampling from p(x_0). But I don't quite understand why one should sample sigma rather than just using a deterministic schedule like other SDE solvers. Does it correspond to some adaptive solvers?

**Summary Of The Paper:**

This paper proposes a faster algorithm, DMCMC, for integrating the reverse SDE/ODE associated with diffusion models. Unlike many existing works, DMCMC samples in the joint space of (x, sigma). Gibbs sampling is adopted for sampling in the joint space. Sampling x | sigma is just ordinary Langevin dynamics, and sampling sigma | x requires additional training of a time prediction network. After MCMC sampling, the algorithm will traverse to a time close to 0. Then, differential equation solvers can be adopted to generate the final sample. DMCMC can be combined with existing solvers, and it can generate high-quality samples in 10~20 NFE.

**Summary Of The Review:**

The paper proposes a novel way to accelerate the sampling of diffusion models. The proposed method can be combined with other approaches, which is an advantage. However, there are still some unclear aspects about the spectrum of the applicability / mechanism of the acceleration behavior.

---

> ### Author Response · Authors · 2022-11-12
> **Response to Reviewer vzaX (Part 3/3)**
>
> **(Continued answer to Comment 3)**
>
> More rigorously, given the same computation budget and the same integration method, integrating over $(\sigma_{\min}, \sigma_n)$ has less truncation error than integrating over $(\sigma_{\min}, \sigma_{\max})$. Computation budget roughly corresponds to the number of discretization points of the integration interval we can use to approximate the reverse-S/ODE. Thus, a shorter interval of integration means we can use smaller step size $h$ during integration, which implies smaller error. For instance, Euler’s method has $O(h^2)$ local error. A more rigorous exposition is given in Chapter 7 of [5].
>
> This justifies how DMCMC can generate better samples than standard diffusion under a fixed computation budget. In other words, DMCMC can use less computation budget than standard diffusion to achieve similar sample quality as standard diffusion, i.e., DMCMC can accelerate sampling.
>
> *But I don't quite understand why one should sample sigma rather than just using a deterministic schedule like other SDE solvers.*
>
> We recall that DMCMC consists of two steps that are executed sequentially: (a) MCMC on $\hat{p}(\mathbf{x},\sigma)$, and (b) denoising MCMC samples by reverse-S/ODE. We also explained how step (a) is crucial for accelerating the generation of clean data. We need to sample sigma, because it is part of MCMC for sampling from $\hat{p}(\mathbf{x},\sigma)$ i.e., a part of step (a) (see Section 4). Given an MCMC sample $(\mathbf{x}\_n,\sigma_n)$ from $\hat{p}(\mathbf{x},\sigma)$ we use a deterministic noise schedule decreasing from $\sigma_n$ to $\sigma_{\min}$ in part (b) to denoise $\mathbf{x}\_n$ into clean data.
>
> *Does it (DMCMC) correspond to some adaptive solver?*
>
> DMCMC does not correspond to some adaptive solver, since we do not propose an adaptive noise decay scheme or an integration scheme for solving reverse-S/ODEs. DMCMC accelerates diffusion sampling by generating better initialization points (initialization points that are better than prior noise samples) for reverse-S/ODE.
>
> **Comment 4. I think it would be better to include a pseudocode in the paper**
>
> We have added a pseudocode for DLG in Appendix C. Moreover, all the hyper-parameters are provided in Appendix A. We have also added a reproducibility statement (Section 7), which contains an anonymized link to our main experiment codes.
>
> **References**
>
> [5] Stoer and Bulisch, Introduction to Numerical Analysis, volume 12, 2002.

---

> ### Author Response · Authors · 2022-11-12
> **Response to Reviewer vzaX (Part 2/3)**
>
> **Comment 3. It is not that intuitive to me how sampling in the joint space can be much faster than annealed MCMC. Doesn't the sampler still require many steps in the x space to mix? It would be better if the authors can better explain the mechanism behind the acceleration behavior. I understand that sampling from annealed distributions is better than directly sampling from p(x_0). But I don't quite understand why one should sample sigma rather than just using a deterministic schedule like other SDE solvers. Does it correspond to some adaptive solvers?**
>
> *Doesn’t the sampler still require many steps in the $\mathbf{x}$ space to mix?*
>
> On the contrary, we find Langevin Gibbs mixes rapidly. In Section 5.1, rightmost panel of Figure 2, we observe that the autocorrelation of image labels vanishes after only a few iterations. If the sampler mixed poorly in the $\mathbf{x}$ space, image labels would have high autocorrelation. For visualization of DLG chains, we refer the Reviewer to Appendix B.1. So, we can say Langevin-Gibbs reliably produces samples from $\hat{p}(\mathbf{x},\sigma)$ even with a small number of steps.
>
> *It is not that intuitive to me how sampling in the joint space can be much faster than annealed MCMC. It would be better if the authors can better explain the mechanism behind the acceleration behavior.*
>
> It is already known [4] that sampling with reverse-S/ODE integrators is faster than annealed Langevin dynamics (ALD). So, the Reviewer’s question boils down to explaining why DMCMC combined with a reverse-S/ODE integrator is faster than using a reverse-S/ODE integrator only. To begin, we note that DMCMC consists of two steps that are executed sequentially: (a) MCMC on $\hat{p}(\mathbf{x},\sigma)$, and (b) denoising MCMC samples by reverse-S/ODE. Since DMCMC without (a) is just standard diffusion, we see the acceleration behavior comes from using $\hat{p}(\mathbf{x},\sigma)$ samples as initial points for reverse-S/ODE. Thus, we need to show how using $\hat{p}(\mathbf{x},\sigma)$ samples as initial points for reverse-S/ODE accelerates image generation.
>
> This proceeds in two steps. We first explain how MCMC produces samples $(\mathbf{x}\_n, \sigma_n)$ from $\hat{p}(\mathbf{x},\sigma)$ such that $\sigma_n$ is significantly smaller than $\sigma_{\max}$. Then, we explain how integrating over $(\sigma_{\min}, \sigma_n)$ is faster than integrating over $(\sigma_{\min}, \sigma_{\max})$ as in standard diffusion.
>
> We first explain how MCMC produces samples $(\mathbf{x}\_n,\sigma_n)$ from $\hat{p}(\mathbf{x}, \sigma)$ such that $\sigma_n$ is significantly smaller than $\sigma_{\max}$.  We observe $\int p_\sigma(\mathbf{x} | \tilde{\mathbf{x}}) p(\tilde{\mathbf{x}}) d\tilde{\mathbf{x}}$ becomes flatter and wider as $\sigma$ increases. This is because $\int p_\sigma(\mathbf{x} | \tilde{\mathbf{x}}) p(\tilde{\mathbf{x}}) d\tilde{\mathbf{x}}$ means we are applying Gaussian smoothing to $p(\mathbf{x})$ with a Gaussian kernel of variance $\sigma^2$. For instance, if $p(\mathbf{x})$ is a normal distribution with variance $\gamma^2$, $\int p_\sigma(\mathbf{x} | \tilde{\mathbf{x}}) p(\tilde{\mathbf{x}}) d\tilde{\mathbf{x}}$ is a normal distribution with variance $\sigma^2 + \gamma^2$. It follows that high density values of $\hat{p}(\mathbf{x},\sigma)$ occur when $\mathbf{x}$ is near the data manifold and $\sigma$ is small. Since MCMC traverses high-probability regions, we can expect $\{\mathbf{x}\_n\}$ will be close to the data manifold and $\{\sigma_n\}$ will be smaller than $\sigma_{\max}$. Indeed, in Appendix F.1, we see that actual $\sigma_n$ values are significantly smaller than $\sigma_{\max} = 50$ in CIFAR10.
>
> Standard diffusion needs to integrate the reverse-S/ODE over the large interval $(\sigma_{\min}, \sigma_{\max})$ to produce clean images. On the other hand, in DMCMC, MCMC produces samples $(\mathbf{x}\_n,\sigma_n)$ from $\hat{p}(\mathbf{x},\sigma)$ such that $\sigma_n$ is significantly smaller than $\sigma_{\max}$. So, in the DMCMC framework, to generate clean images, we only need to integrate the reverse-S/ODE over the small interval $(\sigma_{\min}, \sigma_n)$. This means the cost of integrating over $(\sigma_n, \sigma_{\max})$ vanishes for DMCMC, leading to accelerated image generation. (To be precise, the cost of integrating over $(\sigma_n, \sigma_{\max})$ is replaced by the cost of running MCMC to sample from $\hat{p}(\mathbf{x},\sigma)$, but we observe in Section 5.1 that DLG mixes rapidly, so this cost is negligible.)
>
> **(Answer to Comment 3 continued in Part 3/3)**
>
> **References**
>
> [4] Score-Based Generative Modeling through Stochastic Differential Equations, ICLR, 2021.

---

> > ### Comment · Reviewer_vzaX · 2022-12-07
> > **Thanks for the responses**
> >
> > Thanks the authors for the detailed response. However, I am still somewhat confused about the acceleration behavior.
> >
> > About the intuitive explanation of the acceleration behavior. I admit that the two claimed steps by the authors are both true. But the comparison is not fair.
> >
> > The proposed MCMC has two parts of costs:
> > a) use MCMC to produce (x_n, \sigma_n)
> > b) use solver to integrate over (\sigma_min, \sigma_n).
> >
> > Similarly, we can write the cost of pure solver in two parts:
> > c) integrate over (\sigma_n, \sigma_max).
> > d) integrate over (\sigma_min, \sigma_n).
> >
> > I think the authors are claiming that b < c+d. However, what we want to learn is whether a+b < c+d. Given that b=d.
> >
> > I think what I am expecting is how can we show that a < c. (not talking about experimental results, just more explanation of why MCMC mixes faster for producing a sample x_n).

---

> > > ### Author Response · Authors · 2022-12-07
> > > **Why MCMC mixes faster for producing a sample x_n**
> > >
> > > Simply put, the cost of MCMC is less than the cost of integrating over $(\sigma_n,\sigma_{\max})$ because (1) MCMC chain mixes immediately due to warm start while (2) standard diffusion assumes the denoising distribution is Gaussian, and this assumption generally holds only in the limit of infinitesimal step size. So, the NFE spent in mixing the MCMC chain is less than the NFE spent in integrating over $(\sigma_n,\sigma_{\max})$.
> > >
> > > Warm start is a technique which sets the starting point of MCMC close to a sample from the stationary distribution of MCMC. In fact, if $(\mathbf{x}_0,\sigma_0)$ is exactly a sample from the stationary distribution of MCMC, by the definition of "stationary distribution", $(\mathbf{x}_1,\sigma_1), (\mathbf{x}_2,\sigma_2), \ldots$ will also be samples from the stationary distribution of MCMC. In other words, the chain converges immediately. Theoretical works such as (but not limited to) [5] and [6] tell us warm start is a highly effective technique for accelerating the convergence of MCMC.
> > >
> > > In our work, we set the starting points $(\mathbf{x}_0,\sigma_0)$ of MCMC as points whose distributions are close to the stationary distribution of MCMC (Section 4.1, second paragraph). In Figure 15 in Appendix F.1, we see the samples from the stationary distribution of MCMC have $\sigma_n$ near $0.5$. As described in Algorithm 1 in Appendix C, we generate starting points for MCMC by first generating $\mathbf{x}_0$ from $p(\mathbf{x})$, adding Gaussian noise of standard deviation $0.5$ to $\mathbf{x}_0$, and setting $\sigma_0 = 0.5$. This tells us that the distribution of starting points generated by Algorithm 1 and the stationary distribution of MCMC are very similar. So, it is not surprising the DLG chain mixes almost immediately (e.g., Figure 2).
> > >
> > > On the other hand, integration over $(\sigma_n,\sigma_{\max})$ requires more than just a few NFE. This is because $p(\mathbf{x} \mid \sigma_n)$ is multimodal (recall that $\sigma_n$ is small) while $p(\mathbf{x} \mid \sigma_{\max}) \approx \mathcal{N}(\mathbf{x} \mid \mathbf{0},\sigma_{\max}^2\mathbf{I})$ is unimodal. Specifically, standard diffusion assumes the denoising distributions are Gaussian, and since $p(\mathbf{x} \mid \sigma_n)$ is multimodal while $p(\mathbf{x} \mid \sigma_{\max})$ is unimodal, this assumption holds only in the limit of infinitesimal step size (see Section 3.1 in [7]). Thus, standard diffusion solvers require large NFE to integrate over  $(\sigma_n,\sigma_{\max})$.
> > >
> > > **References**
> > >
> > > [5] Theoretical guarantees for approximate sampling from smooth and log-concave densities, JRSS, 2017.
> > >
> > > [6] Log-concave sampling: Metropolis Hastings algorithms are fast, JMLR, 2019.
> > >
> > > [7] Tackling the Generative Learning Trilemma with Denoising Diffusion GANs, ICLR, 2022.

---

> ### Author Response · Authors · 2022-11-12
> **Response to Reviewer vzaX (Part 1/3)**
>
> **Comment 1. How does the proposed method work with more recent fast solvers, such as DPM-Solver [1] and DEIS [2]? I wonder whether the proposed approach can still help if the solver itself is fast enough.**
>
> We believe our experiments in Section 5.2 can already answer this question. This is because the deterministic integrator of [3], denoted KAR1 in our paper, is a recently proposed (uploaded on arXiv in June 2022) fast solver.
>
> Also, we could not use DPM-Solver and DEIS for the following reasons. We work in the VE diffusion setting, but DPM-Solver does not provide implementation details or code for VE diffusion. In fact, DPM-Solver is only demonstrated in the VP diffusion setting. In the case of DEIS, there is a large performance gap in the VE and VP settings. Specifically, the performance of DEIS deteriorates significantly in the VE setting. In addition, DEIS performs worse than KAR1 in the VE diffusion setting (compare Figure 2 (b) in [3] with Table 10 in [2]). To the best of our knowledge, there is no solver that performs better than KAR1 in the VE diffusion setting with limited NFE.
>
> In Figure 3 of Section 5.2, we observe that DLG still improves the performance of KAR1. In fact, KAR1 combined with DLG beats DPM-Solver (VP) and DEIS (VP) at all NFE and achieves SOTA results in the limited NFE setting (see Review Table 1). We see that DLG improves KAR1 performance on CelebA-HQ-256 (see Figure 4 of Section 5.2) as well. This shows DMCMC can help even if the solver is fast in the first place.
>
> It would be best if we could apply DMCMC to VP diffusion as well. Then, we could accelerate methods such as DPM-Solver or DEIS. However, as discussed in Appendix E.3, combining MCMC with VP diffusion turns out to be a non-trivial task that we believe deserves a paper of its own.
>
> *Review Table 1: Comparison of fast samplers on CIFAR10 (FID) in the limited NFE setting. Number in parenthesis indicates extra or less NFE used. Best numbers are bolded*
>
> | Method | NFE 10 | NFE 20 | NFE 50 |
> |---|---|---|---|
> | DPM-Solver-2 (VP) | 5.28 (+2 NFE) | 3.02 (+4 NFE) | 2.69 (-2 NFE) |
> | DPM-Solver-3 (VP) | 6.03 (+2 NFE) | 2.75 (+4 NFE) | 2.65 (-2 NFE) |
> | DEIS (VP) | 4.17 (+0 NFE) | 2.86 (+0 NFE) | 2.57 (+0 NFE) |
> | DEIS (VE) | 20.89 (+0 NFE) | 16.59 (+0 NFE) | 16.31 (+0 NFE) |
> | KAR1 (VP) | 9.70 (+1 NFE) | 3.23 (+5 NFE) | 2.97 (+1 NFE) |
> | KAR1 (VE) | 14.12 (+1 NFE) | 4.46 (+5 NFE) | 4.1 (+1 NFE) |
> | (Ours) DLG+KAR1 (VE) | $\mathbf{3.86}$ (+0.1 NFE) | $\mathbf{2.63}$ (+0.1 NFE) | $\mathbf{2.45}$ (-0.9 NFE) |
>
> **Comment 2. The noise predictor network is a trained distribution rather than the true conditional p(sigma | x). With such an approximate conditional, MCMC can no longer guarantee to converge to the true stationary distribution (so rigorously it is not a Gibbs sampling algorithm). I wonder to what extent does this approximation affect the convergence / sample quality.**
>
> We note that all recent diffusion models rely on trained score models to solve the reverse-S/ODE. Hence, similar to what the Reviewer has said, diffusion models can no longer be guaranteed to generate samples from the true data distribution. Despite this theoretical drawback, diffusion models have shown excellent performance in data generation. So, in an analogous manner, we do not believe using a conditional approximated conditional is a drawback of DMCMC. Indeed, experiment results show using an approximate conditional does not harm the convergence of MCMC or the final sample quality.
>
> *Accuracy of approximate conditional*
>
> In Appendix F.1, we perform an in-depth analysis of DLG $\sigma$ sequence, and we see that the noise predictor network accurately predicts the optimal noise scale given $\mathbf{x}$ such that the MCMC chain does not diverge.
>
> *Convergence*
>
> In Section 5.1, even though we use the noise predictor network to run DLG on a mixture of Gaussians (MoG), we see in Figure 2 of Section 5.1 that the samples have perfect mode coverage and accurate label distribution. If DLG had not converged to the MoG distribution, we would not have seen such mode coverage and correct label distribution. The rightmost panel of Figure 2 shows that autocorrelation decays rapidly, and this implies that the MCMC chain converges fast.
>
> *Sample quality*
>
> As for the sample quality, we believe the results of Section 5.2 speak for themselves. DLG is capable of producing high-quality, diverse samples rapidly, even though we use an approximate conditional. So, we see that this approximation does not harm sample quality.
>
> **References**
>
> [1] DPM-Solver: A Fast ODE Solver for Diffusion Probabilistic Model Sampling in Around 10 Steps, NeurIPS, 2022.
>
> [2] Fast Sampling of Diffusion Models with Exponential Integrator, arXiv, 2022.
>
> [3] Elucidating the Design Space of Diffusion-Based Generative Models, NeurIPS, 2022.

---

> ### Author Response · Authors · 2022-11-16
> **Do our revisions and responses answer your concerns and questions?**
>
> Dear Reviewer vzaX,
>
> We thank the Reviewer for the constructive comments. As the end of the paper revision period is approaching, we would like to ask whether our paper revisions and responses have addressed your concerns and questions adequately. If not, we would be happy to discuss and update our paper further.
>
> Regards,
>
> the authors.

---

> > ### Author Response · Authors · 2022-12-05
> > **We are looking forward to your feedback**
> >
> > Dear Reviewer vzaX,
> >
> > As there is only one week left until the end of the author-reviewer discussion stage, we would like to ask again whether our paper revisions and responses have addressed your concerns and questions adequately. If not, we would be happy to discuss further.
> >
> > Regards,
> >
> > the authors.

---

### Official Review · Reviewer_uZQw · 2022-10-24

**Confidence:** 4
**Correctness:** 3
**Technical Novelty And Significance:** 2
**Empirical Novelty And Significance:** 3
**Recommendation:** 5

**Clarity, Quality, Novelty And Reproducibility:**

Paper is clearly written. I think there is a bit of uncertainty regarding what really happens with the sequence $\sigma$ so the methodology is not perfectly clear.

The work is coherent and the methodology is a nice idea.

It seems to be novel although I really think that a similar idea was introduced in [1] and should be discussed.

The authors give experimental details.

[1] Noise Estimation for Generative Diffusion Models - San Roman, Nachmani, Wolf


**Details Of Ethics Concerns:**

No ethical concern.

**Strength And Weaknesses:**

STRENGTHS:
* I think the ideas of the paper are clearly explained. I like the idea of extending the state space to also sample from the parameter s$\sigma$. I think this is an elegant idea that could lead to further developments.
* The experimental results seem to confirm that using the proposed method Denoising Langevin Gibbs (DLG) brings an advantage to existing sampling techniques.

WEAKNESSES:
* At the core level I think there is something weird with the strategy. Classical denoising diffusion models aim at denoising the image. They start with a Gaussian random variable and then go backward in time to reach the initial distribution. In that case the $\sigma$ decrease towards $0$ which makes sense as we are trying to go back to the data distribution. But here this is quite different as we are also sampling from $p(\sigma|x)$. At the end of the day (if things were perfect) we should sample from $p(x, \sigma)$. Hence, the noise should travel to the whole interval $[\sigma_{\mathrm{min}}, \sigma_{\mathrm{max}}]$ and not be stuck at $\sigma_{\mathrm{min}}$. Being stuck near $\sigma_{\mathrm{min}}$ has the advantage that then the samples are perceptually good but then we do not explore the whole distribution. It would be incredibly useful and I think key to the paper if the authors could provide the evolution of the sequence of $\sigma$. If this sequence is decreasing only then we are not exploring the distribution and the claim that we are doing a Markov chain in the augmented space is misleading. Maybe this is due to the fact that we use a classifier to learn the noise level $\sigma$. In any case, I think that the sampling on $\sigma$ which is the real methodological contribution of the paper is not investigated enough by the authors.
* Why do the authors focus on the Variance Exploding SDE? In the literature it has been shown that usually VP-SDE (which can be connected to DDPM) are more stable and yield better results than VE-SDE. This is because VE-SDE is just a Brownian motion whereas VP-SDE is a Ornstein-Uhlenbeck process (for which the variance does not explode and converge to the dimension). I am pretty sure that the analysis still holds in that case.
* I think that the authors missed a few references trying to predict the noise level in the sampling. For instance [1] also learned the covariance in DDPM models. In [2] the authors learn the noise level. The method seems very similar to what is introduced and is not discussed in the paper.
* I think that the claims on SOTA for FID for Celeba256x256 as some methods seem to achieve 3.25 see https://paperswithcode.com/sota/image-generation-on-celeba-256x256

OTHER QUESTIONS/COMMENTS:
* I don't really understand the discussion at the end of Section 3.2.
* In Equation (15) the authors seem to be using the Langevin dynamics targetting (the approximate) noised target distribution. This corresponds to the corrector step in [3]. Empirical analysis in [3] shows that using only the corrector step and not the predictor-corrector (or only predictor) step was very detrimental to the quality of the sampling. However it does not seem to affect the results of the present work. Could the authors give more details about this?

[1] Improved Denoising Diffusion Probabilistic Models - Nichol, Dhariwal
[2] Noise Estimation for Generative Diffusion Models - San Roman, Nachmani, Wolf
[3] Score-Based Generative Modeling through Stochastic Differential Equations - Song, Sohl-Dickstein, Kingma, Kumar, Ermon, Poole

**Summary Of The Paper:**

In this paper the authors propose to extend the state space of diffusion models to answer the following question: how much should we sample at a given noise level? The way they proceed is as follows. First, they put some prior on the space of noise level $p(\sigma)$ (in practice this distribution is chosen so that $p(\sigma) \propto 1/\sigma$ to put more emphasis on the distribution with small noise. This is because we ultimately want to sample from the original distribution  (without noise). Then we can easily write the joint distribution on the space of image and noise levels by using $p(x|\sigma)p(\sigma)$. Then we sample using a Gibbs procedure which can be described as follows:
* Given a noise level we can approximately sample from $p(x|\sigma)$ leveraging diffusion models. In particular, we use an Unadjusted Langevin Dynamics targetting $p(x|\sigma)$ where the gradient of the logarithm of this density is given by the learned score of the diffusion model. This part is really similar to a diffusion model
* Second we need to sample from $p(\sigma|x)$. To do so the authors train a neural network to predict the noise level. In fact even though we need to learn a probability distribution the authors found that just outputing one value of $\sigma$ corresponding to the mode of the distribution worked well.
The authors then alternate between the two steps. They illustrate their method on CelebA-HQ 256 and CIFAR 10.

**Summary Of The Review:**

I think this work is an interesting contribution to the field.
However, as of now I think that there are a few elements missing:
* a discussion of the literature to assess the novelty of the idea
* a discussion of some methodological choices that seem to be quite non-classical (VE-SDE, corrector only step)
* verify claim on SOTA for Celeba256.
* most importantly, a study of the evolution of $\sigma$ to assess if we are really sampling from a Markov chain.
For these reasons, I think that as of now reasons to reject the paper outweight reasons to accept it.

---

> ### Author Response · Authors · 2022-11-12
> **Response to Reviewer uZQw (Part 5/5)**
>
> **Comment 6. In Equation (15) the authors seem to be using the Langevin dynamics targeting (the approximate) noised target distribution. This corresponds to the corrector step in [7]. Empirical analysis in [7] shows that using only the corrector step and not the predictor-corrector (or only predictor) step was very detrimental to the quality of the sampling. However, it does not seem to affect the results of the present work. Could the authors give more details about this?**
>
> This is because we apply denoising via reverse-S/ODE integrators to MCMC samples of $\hat{p}(\mathbf{x},\sigma)$, as mentioned in Section 3. MCMC part of DMCMC by itself is unable to produce clean images (see the bottom right part of Figure 1. MCMC images in the gray box have small noise levels compared to $\sigma_{\max}$, but the perceptual quality of the samples is still poor. Denoising via reverse-S/ODE integrators produces clean samples, below the gray box). In Appendix F.2, we indeed observe that removing the denoising step significantly harms the performance of DMCMC.
>
> We also note that “predictor-corrector” is a collection of algorithms for integrating differential equations (https://en.wikipedia.org/wiki/Predictor%E2%80%93corrector_method). However, in the first construction step of DMCMC, we do not solve any differential equations. The goal of the first construction step Section 3.1 is to generate samples from $\hat{p}(\mathbf{x},\sigma)$. So, the concept of “predictor-corrector” is not directly applicable to the MCMC part of DMCMC.
>
> What we believe the Reviewer meant to say is that (please correct us if we are wrong) we could construct a diffusion process whose reverse-S/ODE converges to $\hat{p}(\mathbf{x},\sigma)$. Then, we can incorporate “predictor” steps into sampling of $\hat{p}(\mathbf{x},\sigma)$. However, this would require training a score network for $\hat{p}(\mathbf{x},\sigma)$. This beats the purpose of DMCMC, which is efficient data generation with pre-trained score models. Moreover, we see in Section 5.1 that Langevin Gibbs converges very rapidly (autocorrelation vanishes after only a few iterations), so it samples from $\hat{p}(\mathbf{x},\sigma)$ efficiently.
>
> **References**
>
> [7] Score-Based Generative Modeling through Stochastic Differential Equations, ICLR, 2021.

---

> ### Author Response · Authors · 2022-11-12
> **Response to Reviewer uZQw (Part 4/5)**
>
> **Comment 5. I don't really understand the discussion at the end of Section 3.2.**
>
> The following content has been added as Appendix E.1 in our revised paper.
>
> The discussion at the end of Section 3.2 describes why DMCMC combined with a reverse-S/ODE integrator is faster than using a reverse-S/ODE integrator alone. Here, we provide a more detailed explanation of the acceleration phenomenon. To begin, we note that DMCMC consists of two steps that are executed sequentially: (a) MCMC on $\hat{p}(\mathbf{x},\sigma)$, and (b) denoising MCMC samples by reverse-S/ODE. Since DMCMC without (a) is just standard diffusion, we see the acceleration behavior comes from using $\hat{p}(\mathbf{x},\sigma)$ samples as initial points for reverse-S/ODE. Thus, we need to show how using $\hat{p}(\mathbf{x},\sigma)$ samples as initial points for reverse-S/ODE accelerates image generation.
>
> This proceeds in two steps. We first explain how MCMC produces samples $(\mathbf{x}\_n, \sigma_n)$ from $\hat{p}(\mathbf{x},\sigma)$ such that $\sigma_n$ is significantly smaller than $\sigma_{\max}$. Then, we explain how integrating over $(\sigma_{\min}, \sigma_n)$ is faster than integrating over $(\sigma_{\min}, \sigma_{\max})$ as in standard diffusion.
>
> We first explain how MCMC produces samples $(\mathbf{x}\_n,\sigma_n)$ from $\hat{p}(\mathbf{x}, \sigma)$ such that $\sigma_n$ is significantly smaller than $\sigma_{\max}$.  We observe $\int p_\sigma(\mathbf{x} | \tilde{\mathbf{x}}) p(\tilde{\mathbf{x}}) d\tilde{\mathbf{x}}$ becomes flatter and wider as $\sigma$ increases. This is because $\int p_\sigma(\mathbf{x} | \tilde{\mathbf{x}}) p(\tilde{\mathbf{x}}) d\tilde{\mathbf{x}}$ means we are applying Gaussian smoothing to $p(\mathbf{x})$ with a Gaussian kernel of variance $\sigma^2$. For instance, if $p(\mathbf{x})$ is a normal distribution with variance $\gamma^2$, $\int p_\sigma(\mathbf{x} | \tilde{\mathbf{x}}) p(\tilde{\mathbf{x}}) d\tilde{\mathbf{x}}$ is a normal distribution with variance $\sigma^2 + \gamma^2$. It follows that high density values of $\hat{p}(\mathbf{x},\sigma)$ occur when $\mathbf{x}$ is near the data manifold and $\sigma$ is small. Since MCMC traverses high-probability regions, we can expect $\{\mathbf{x}\_n\}$ will be close to the data manifold and $\{\sigma_n\}$ will be smaller than $\sigma_{\max}$. Indeed, in Appendix F.1, we see that actual $\sigma_n$ values are significantly smaller than $\sigma_{\max} = 50$ in CIFAR10.
>
> Standard diffusion needs to integrate the reverse-S/ODE over the large interval $(\sigma_{\min}, \sigma_{\max})$ to produce clean images. On the other hand, in DMCMC, MCMC produces samples $(\mathbf{x}\_n,\sigma_n)$ from $\hat{p}(\mathbf{x},\sigma)$ such that $\sigma_n$ is significantly smaller than $\sigma_{\max}$. So, in the DMCMC framework, to generate clean images, we only need to integrate the reverse-S/ODE over the small interval $(\sigma_{\min}, \sigma_n)$. This means the cost of integrating over $(\sigma_n, \sigma_{\max})$ vanishes for DMCMC, leading to accelerated image generation. (To be precise, the cost of integrating over $(\sigma_n, \sigma_{\max})$ is replaced by the cost of running MCMC to sample from $\hat{p}(\mathbf{x},\sigma)$, but we observe in Section 5.1 that DLG mixes rapidly, so this cost is negligible.)
>
> More rigorously, given the same computation budget and the same integration method, integrating over $(\sigma_{\min}, \sigma_n)$ has less truncation error than integrating over $(\sigma_{\min}, \sigma_{\max})$. Computation budget roughly corresponds to the number of discretization points of the integration interval we can use to approximate the reverse-S/ODE. Thus, a shorter interval of integration means we can use smaller step size $h$ during integration, which implies smaller error. For instance, Euler’s method has $O(h^2)$ local error. A more rigorous exposition is given in Chapter 7 of [6].
>
> This justifies how DMCMC can generate better samples than standard diffusion under a fixed computation budget. In other words, DMCMC can use less computation budget than standard diffusion to achieve similar sample quality as standard diffusion, i.e., DMCMC can accelerate sampling.
>
> **References**
>
> [6] Stoer and Bulisch, Introduction to Numerical Analysis, volume 12, 2002.

---

> ### Author Response · Authors · 2022-11-12
> **Response to Reviewer uZQw (Part 3/5)**
>
> **(Continued answer to Comment 2)**
>
> Due to these pathologies of VP diffusion, what we observed happening in practice was that Langevin Gibbs would repeat the following behavior: try to approach the image manifold in the $\mathbf{x}$ update step, step off the high-density path leading to the data manifold in the $\mathbf{x}$ update step, and get absorbed into the prior noise distribution. Indeed, in Figure 14 of Appendix E.3 we observe $\alpha$ values ($\alpha$ is the mixing coefficient of data with a prior noise sample. $\alpha = 0$ means $\mathbf{x}$ is a prior noise sample, and $\alpha = 1$ means $\mathbf{x}$ is a data sample.) oscillating around zero. This implies $\mathbf{x}\_n$ are essentially prior noise samples, so using $\{(\mathbf{x}\_n,\alpha_n)\}$ as initial points for the reverse-S/ODE has no practical benefit.
>
> This type of problem with narrow, high-density regions often plagues MCMC. For a toy example, see Figure 3 of [3]. We postulate that a better MCMC sampling scheme could enable us to run DMCMC under the VP diffusion framework. For instance, one could use MCMC with rejection steps that would reject points that step off high-density regions, or incorporate Riemannian manifold structure into the sampling scheme. However, we believe this is a topic for future work.
>
> **Comment 3. I think that the authors missed a few references trying to predict the noise level in the sampling. For instance [4] also learned the covariance in DDPM models. In [5] the authors learn the noise level. The method seems very similar to what is introduced and is not discussed in the paper.**
>
> We thank the Reviewer for the missing references. The following discussion is added as Appendix D.2 in the revised paper. Although both works employed a neural net to predict the noise level / diffusion time for a given data input, we emphasize that the previous studies pointed out by Reviewer uZQw do not overlap with the contributions of DMCMC. The main contribution of our paper is not the noise prediction network itself. The novelty of DMCMC arises from how we use the noise prediction network.
>
> On a high level, the main message of our paper is that we can significantly improve diffusion sampling by choosing better initial points for denoising. A noise predictor network was used in the process of finding initial points. On the other hand, the focus of the mentioned works is on improving the denoising process with a noise predictor network. So, DMCMC and the mentioned works contribute to orthogonal aspects of diffusion sampling.
>
> Let us elaborate. We first note that diffusion model sampling can be broken down into two parts: (part 1) generating an initialization point, and (part 2) integrating the reverse-S/ODE to generate data from the initialization point.
>
> [4] and [5] use the noise prediction network to accelerate integration of the reverse-S/ODE (improve part 2). This is achieved by learning the covariance of the reverse distribution [4] or by adjusting the noise schedule with a neural net [5].
>
> In DMCMC, MCMC generates initialization points for reverse-S/ODE that lie near the image manifold (improve part 1). This is possible because MCMC runs in $\mathcal{X} \times \mathcal{S}$ by adaptively updating the noise level with a noise prediction network. This leads to acceleration, as it is easier for reverse-S/ODE integrators to generate clean data from points near the data manifold than from prior noise distribution samples (e.g., Gaussian noise). Hence, our paper proposes an acceleration approach entirely different from those of and [4] and [5].
>
> In fact, DMCMC does not resemble [4] and [5] even from the perspective of implementation. Noise predictors in [4] and [5] are used during reverse-S/ODE integration, and noise predictor in DMCMC is used before reverse-S/ODE integration, as a sub-step of MCMC.
>
> **Comment 4. I think that the claims on SOTA for FID for Celeba256x256 as some methods seem to achieve 3.25 see https://paperswithcode.com/sota/image-generation-on-celeba-256x256**
>
> We intended to say that DMCMC achieves SOTA among score-based models. On the mentioned site, all methods that achieve around 3~4 FID are GANs. We acknowledge that our writing could be misleading. Hence, we have modified our abstract and introduction accordingly.
>
> **References**
>
> [3] Introducing an Explicit Symplectic Integration Scheme for Riemannian Manifold Hamiltonian Monte Carlo, arXiv, 2019.
>
> [4] Improved Denoising Diffusion Probabilistic Models, ICML, 2021.
>
> [5] Noise Estimation for Generative Diffusion Models, arXiv, 2021.

---

> ### Author Response · Authors · 2022-11-12
> **Response to Reviewer uZQw (Part 2/5)**
>
> **Comment 2. Why do the authors focus on the Variance Exploding SDE? In the literature it has been shown that usually VP-SDE (which can be connected to DDPM) are more stable and yield better results than VE-SDE. This is because VE-SDE is just a Brownian motion whereas VP-SDE is a Ornstein-Uhlenbeck process (for which the variance does not explode and converge to the dimension).**
>
> It is true that the VP setting could be more stable than the VE setting, and that is why some recent solvers work in the VP setting. For instance, DPM-Solver [1] only provides experiment results in the VP setting despite, the fact that DPM-Solver can be applied to the VE setting as well. In the case of DEIS [2], there is a large performance gap in the VE and VP settings. Specifically, the performance of DEIS deteriorates significantly in the VE setting.
>
> We used VE diffusion because, from the perspective of generating better initialization points via MCMC, VE setting was better than VP setting. We believe the benefits outweigh the downsides, since, as shown in Review Table 1 below, DLG beats all recent fast solvers regardless of whether that fast solver works in the VP setting or the VE setting. We conjecture that stability does not have a large influence on DMCMC because the reverse-S/ODE initialization points generated by MCMC are close to the image manifold, such that the variance of initialization points is small compared to the variance of prior noise. This is possibly why DMCMC improves the performance lower bound for some deterministic integrators (Section 5.2).
>
> *Review Table 1: Comparison of fast samplers on CIFAR10 (FID) in the limited NFE setting. Number in parenthesis indicates extra or less NFE used. Best numbers are bolded*
>
> | Method | NFE 10 | NFE 20 | NFE 50 |
> |---|---|---|---|
> | DPM-Solver-2 (VP) | 5.28 (+2 NFE) | 3.02 (+4 NFE) | 2.69 (-2 NFE) |
> | DPM-Solver-3 (VP) | 6.03 (+2 NFE) | 2.75 (+4 NFE) | 2.65 (-2 NFE) |
> | DEIS (VP) | 4.17 (+0 NFE) | 2.86 (+0 NFE) | 2.57 (+0 NFE) |
> | DEIS (VE) | 20.89 (+0 NFE) | 16.59 (+0 NFE) | 16.31 (+0 NFE) |
> | KAR1 (VP) | 9.70 (+1 NFE) | 3.23 (+5 NFE) | 2.97 (+1 NFE) |
> | KAR1 (VE) | 14.12 (+1 NFE) | 4.46 (+5 NFE) | 4.1 (+1 NFE) |
> | (Ours) DLG+KAR1 (VE) | $\mathbf{3.86}$ (+0.1 NFE) | $\mathbf{2.63}$ (+0.1 NFE) | $\mathbf{2.45}$ (-0.9 NFE) |
>
> We also give a detailed explanation of why DMCMC is more compatible with VE. Concretely, we observed two differences between VE diffusion and VP diffusion that made MCMC with VP diffusion difficult.
>
> Side note: When we mention the support of a distribution, we mean the set of points where the numerical value (value represented on a computer) of the density is nonzero. This is because, even when a distribution theoretically has nonzero density everywhere, it is possible that the numerical value of the density vanishes outside some bounded region.
>
> In VE diffusion, the support of $\int p_\sigma(\mathbf{x} \mid \tilde{\mathbf{x}}) p(\tilde{\mathbf{x}}) d\tilde{\mathbf{x}}$ grows wider as $\sigma$ increases. Thus, wherever the current MCMC $\mathbf{x}$ iterate is, there always is some noise level $\sigma$ such that $\mathbf{x}$ is in the support of $\int p_\sigma(\mathbf{x} \mid \tilde{\mathbf{x}}) p(\tilde{\mathbf{x}}) d\tilde{\mathbf{x}}$ so the score function provides a meaningful (i.e., non-zero) direction. The noise classification network $q_\phi$ predicts this optimal noise level $\sigma$ for $\mathbf{x}$, as shown in Appendix F.1.
>
> On the other hand, in the VP diffusion framework, the prior noise distribution (standard normal distribution) has finite support. To be precise, the standard normal distribution has non-zero density everywhere, but numerically, density becomes zero at points sufficiently far from the origin. As the data distribution also has finite support, all intermediate distributions $p_t(\mathbf{x})$  of VP diffusion have finite support as well. Since MCMC takes random steps, it is possible for an MCMC iterate to land at a point where no intermediate distribution $p_t(\mathbf{x})$ of VP diffusion provides meaningful density or gradient. Even worse, the mean of the corruption process of VP diffusion shifts in the process of changing data samples to zero mean standard normal samples. So, high-density paths between prior noise samples and data samples can become very narrow in high-dimensional scenarios.
>
> **(Answer to Comment 2 continued in Part 3/5)**
>
> **References**
>
> [1] DPM-Solver: A Fast ODE Solver for Diffusion Probabilistic Model Sampling in Around 10 Steps, NeurIPS, 2022.
>
> [2] Fast Sampling of Diffusion Models with Exponential Integrator, arXiv, 2022.

---

> ### Author Response · Authors · 2022-11-12
> **Response to Reviewer uZQw (Part 1/5)**
>
> **Comment 1. It would be incredibly useful and I think key to the paper if the authors could provide the evolution of the sequence of $\sigma$.**
>
> As a response to Reviewer uZQw’s comment, we provide analyses of $\sigma$ evolution during Langevin Gibbs. In Figure 15 in Appendix F.1, we have visualized Langevin Gibbs trajectories of $\sigma$ in the MoG setting with 1k modes at CIFAR10 images (this setting decouples sampler behavior from score function approximation error) and the score network setting. We indeed observe that $\sigma$ moves up and down, allowing $\mathbf{x}$ to travel between disjoint modes of the distribution. Moreover, in Figure 2 of Section 5.1, $\mathbf{x}$ samples cover all 1k modes of the MoG. This experimentally proves $\mathbf{x}$ sequence of DLG is exploring the entire image distribution.
>
> In the MoG setting, let us define $\delta_n$ as the distance of $\mathbf{x}\_n$ to the closest mode in the MoG. In Figure 16 in Appendix F.1, we see $\sigma_n \sim q_\phi(\sigma \mid \mathbf{x}\_n)$ is almost identical to $\delta_n / \sqrt{d}$, where $d$ is the dimension of $\mathbf{x}\_n$. This is reasonable, as the Gaussian annulus theorem tells us samples from a high-dimensional Gaussian distribution of mean $\mu$ and variance $\sigma^2$ come from a shell of radius $\sigma \sqrt{d}$ centered at $\mu$.
>
> Specifically, due to the Gaussian annulus theorem, the samples of $\int p_\sigma(\mathbf{x} \mid \tilde{\mathbf{x}}) p(\tilde{\mathbf{x}}) d\tilde{\mathbf{x}}$ are most likely to come from a shell of radius $\sigma \sqrt{d}$ centered around the image manifold. (See Figure 2 (a) in the paper by https://arxiv.org/pdf/2206.00941.pdf) Then, given certain $\mathbf{x}$ which is distance $\delta$ from the image manifold, we can intuitively argue that optimal $\sigma$ for $\mathbf{x}$, i.e., $\sigma$ of highest likelihood under $p(\sigma \mid \mathbf{x})$, can be determined by equating $\delta = \sigma \sqrt{d}$ such that $\sigma = \delta / \sqrt{d}$. In the MoG case, this $\delta$ is approximated by the distance of $\mathbf{x}$ to the closest mode in the MoG. Since the $\sigma$ values predicted by the noise classifier agree with approximated $\delta / \sqrt{d}$, we can see that the noise classifier is a good approximation of $p(\sigma \mid \mathbf{x})$.

---

> ### Author Response · Authors · 2022-11-16
> **Do our revisions and responses answer your concerns and questions?**
>
> Dear Reviewer uZQw,
>
> We thank the Reviewer for the constructive comments. As the end of the paper revision period is approaching, we would like to ask whether our paper revisions and responses have addressed your concerns and questions adequately. If not, we would be happy to discuss and update our paper further.
>
> Regards,
>
> the authors.

---

> > ### Author Response · Authors · 2022-12-05
> > **We are looking forward to your feedback**
> >
> > Dear Reviewer uZQw,
> >
> > As there is only one week left until the end of the author-reviewer discussion stage, we would like to ask again whether our paper revisions and responses have addressed your concerns and questions adequately. If not, we would be happy to discuss further.
> >
> > Regards,
> >
> > the authors.

---

### Official Review · Reviewer_7H8V · 2022-10-30

**Confidence:** 3
**Correctness:** 3
**Technical Novelty And Significance:** 2
**Empirical Novelty And Significance:** 2
**Recommendation:** 6

**Clarity, Quality, Novelty And Reproducibility:**

Overall the paper is easy to follow and writing is Ok. The novelty is limited (see my comments above). The code is not provided, as the paper requires quite some tuning, the reproducibility might be an issue.


**Strength And Weaknesses:**

Strength:

- Improving the speed of the reverse process in denoising diffusion models is an important topic.
- The proposed method is able to improve the efficiency, through the experimental justification.

Weakness:

- The novelty is somewhat limited, given the existing predictor-corrector algorithm. Using MCMC to obtain samples from p_t(x) is already done in Song et.al, 2021b. This paper directly uses MCMC to sample from p_t(x), which is a surrogate that has a trade-off between quality and the speed. But overall the technical contribution is limited as it is a relative straightforward extension of existing technique.

- There might be a potential issue for the first stage (i.e., using MCMC for p_t). The score function required for MCMC is approximated using the learned score function. It would be fine if the sample x is roughly around the high density region of p_t(x), however the initial samples fed to the score function might be quite far away. These would be the out-of-distribution samples for the neural parameterized score function, where the neural network may not be able to extrapolate that well.

- Paper [1] is included but is not directly mentioned or compared. The authors are encouraged to provide more experimental comparison with [1].

Reference:

[1] DPM-Solver: A Fast ODE Solver for Diffusion Probabilistic Model Sampling in Around 10 Steps, Lu et.al.



**Summary Of The Paper:**

This paper aims at speeding up the reverse simulation process of score based denoising diffusion models through SDE. The main framework consists of two parts, where the first part is an MCMC based sampler that samples (t, x_t ~ p_t) jointly, where p_t is the marginal distribution of the forward process till time t. The second part is the standard reverse ODE simulation, where the simulation starts from this sampled time t and corresponding location x_t, and ends at t=0 to obtain the final sample. The main contribution comes from the first part where MCMC is implemented via Gibbs and stochastic gradient langevin with learned score function as the approximation. Experiments show that it can save the computation, while also improving several ODE-solver based approaches.


**Summary Of The Review:**

Overall this paper proposed a practical approach for speeding up the reverse process sampling of denoising diffusion models. The main concern comes from the novelty and technical correctness of the assumption/approximation. The experiment section can potentially be improved with more baseline comparison.

====
updated after the authors' response and more experimental details.

---

> ### Author Response · Authors · 2022-11-12
> **Response to Reviewer 7H8V (Part 2/2)**
>
> **Comment 2. There might be a potential issue for the first stage (i.e., using MCMC for p_t). The score function required for MCMC is approximated using the learned score function. It would be fine if the sample x is roughly around the high-density region of p_t(x), however the initial samples fed to the score function might be quite far away. These would be the out-of-distribution samples for the neural parameterized score function, where the neural network may not be able to extrapolate that well.**
>
> That is one of the reasons why we work in the VE diffusion framework. In the VE diffusion framework, the support of $\int p_\sigma(\mathbf{x} | \tilde{\mathbf{x}}) p(\tilde{\mathbf{x}}) d\tilde{\mathbf{x}}$ grows wider and wider as $\sigma \rightarrow \infty$. This indicates that for any given $\mathbf{x}$, there is always some noise level $\sigma$ such that $\mathbf{x}$ is in the support of $\int p_\sigma(\mathbf{x} | \tilde{\mathbf{x}}) p(\tilde{\mathbf{x}}) d\tilde{\mathbf{x}}$. Thus, the noise conditional score network conditioned on that noise level $\sigma$ will provide an accurate score estimate for $\mathbf{x}$.
>
> The noise classification network $q_\phi$ predicts this optimal noise level $\sigma$ for $\mathbf{x}$, as demonstrated in Appendix F.1. Hence, the issue of out-of-distribution (OOD) inputs for the score function is not such a critical problem. Consistent gains in performance, as demonstrated in Section 5.2, supports this claim. If the initial samples provided by DMCMC were OOD samples for the score function, DMCMC would not be able to generate diverse, high-quality images.
>
> Side note: When we mention the support of a distribution, we mean the set of points where the numerical value (value represented on a computer) of the density is nonzero. This is because, even when a distribution theoretically has nonzero density everywhere, it is possible that the numerical value of the density vanishes outside some bounded region.
>
> **Comment 3. Paper DPM-Solver [2] is included but is not directly mentioned or compared. The authors are encouraged to provide more experimental comparison with DPM-Solver.**
>
> In the table below, we compare DMCMC with not only DPM-Solver [2], but also other fast diffusion samplers such as DEIS [3] and KAR1 [4] in the limited NFE setting. We see DLG beats all fast solvers, regardless of whether that fast solver works in the VP setting or the VE setting. This table is added as Table 1 in Section 5.2 in the revised version of our paper.
>
> *Review Table 2: Comparison of fast samplers on CIFAR10 (FID) in the limited NFE setting. Number in parenthesis indicates extra or less NFE used. Best numbers are bolded*
>
> | Method | NFE 10 | NFE 20 | NFE 50 |
> |---|---|---|---|
> | DPM-Solver-2 (VP) | 5.28 (+2 NFE) | 3.02 (+4 NFE) | 2.69 (-2 NFE) |
> | DPM-Solver-3 (VP) | 6.03 (+2 NFE) | 2.75 (+4 NFE) | 2.65 (-2 NFE) |
> | DEIS (VP) | 4.17 (+0 NFE) | 2.86 (+0 NFE) | 2.57 (+0 NFE) |
> | DEIS (VE) | 20.89 (+0 NFE) | 16.59 (+0 NFE) | 16.31 (+0 NFE) |
> | KAR1 (VP) | 9.70 (+1 NFE) | 3.23 (+5 NFE) | 2.97 (+1 NFE) |
> | KAR1 (VE) | 14.12 (+1 NFE) | 4.46 (+5 NFE) | 4.1 (+1 NFE) |
> | (Ours) DLG+KAR1 (VE) | $\mathbf{3.86}$ (+0.1 NFE) | $\mathbf{2.63}$ (+0.1 NFE) | $\mathbf{2.45}$ (-0.9 NFE) |
>
> **Comment 4. Reproducibility might be an issue.**
>
> We have added a pseudocode for DLG in Appendix C. Moreover, all the hyper-parameters are provided in Appendix A. We have also added a reproducibility statement (Section 7), which contains an anonymized link to our main experiment codes.
>
> **References**
>
> [2] DPM-Solver: A Fast ODE Solver for Diffusion Probabilistic Model Sampling in Around 10 Steps, NeurIPS, 2022.
>
> [3] Fast Sampling of Diffusion Models with Exponential Integrator, arXiv, 2022.
>
> [4] Elucidating the Design Space of Diffusion-Based Generative Models, NeurIPS, 2022.

---

> ### Author Response · Authors · 2022-11-12
> **Response to Reviewer 7H8V (Part 1/2)**
>
> **Comment 1. The novelty is somewhat limited, given the existing predictor-corrector algorithm. Using MCMC to obtain samples from p_t(x) is already done in Song et al., 2021b. This paper directly uses MCMC to sample from p_t(x), which is a surrogate that has a trade-off between quality and the speed. But overall the technical contribution is limited as it is a relative straightforward extension of existing technique.**
>
> As both DMCMC and the predictor-corrector (PC) algorithm [1] rely on MCMC, one may ask whether DMCMC is a straightforward extension of PC. However, we assure the Reviewer that DMCMC is not a trivial generalization of PC. (The following discussion has been added as Appendix D.1 in our revised paper.)
>
> On a high level, the main message of our paper is that we can significantly improve diffusion sampling by choosing better initial points for the denoising process. This claim was verified through extensive experiments. On the other hand, the focus of the corrector step of PC is on improving the denoising process by refining the predictor / integrator steps. So, DMCMC and PC contribute to orthogonal aspects of diffusion sampling. Indeed, the improvements offered by the MCMC corrector step are rather marginal (see Table 1 in [1]) compared to the improvements offered by DMCMC (see Figure 3 in Section 5.2 in our paper).
>
> On a low level, we can distinguish DMCMC from PC on three levels: theoretical, implementation-wise, and empirical.
>
> **Theoretical**
>
> Theoretically, MCMC in DMCMC and MCMC in PC reduce truncation error (error arising from numerically integrating reverse-S/ODE) in entirely distinct ways. We observe that diffusion model sampling can be broken down into two parts: (part 1) generating an initialization point, and (part 2) integrating the reverse-S/ODE starting from the initialization point to generate clean data. MCMC in PC aims to improve part 2 whereas MCMC in DMCMC aims to improve part 1.
>
> In PC, MCMC is used as a corrector. That is, MCMC corrects the distributions of intermediate points during integration of the reverse-S/ODE (part 2). That is why PC sampling proceeds by alternating between taking a predictor step and running Langevin dynamics at a fixed noise level.
>
> In DMCMC, MCMC generates initialization points for reverse-S/ODE (part 1) that lie near the image manifold. This is possible because MCMC runs in the augmented space $\mathcal{X} \times \mathcal{S}$ by adaptively updating the noise level. This leads to reduced truncation error, or equivalently, acceleration, as it is easier for reverse-S/ODE integrators to generate clean data from points near the data manifold than from prior noise distribution samples (e.g., Gaussian noise). For a more detailed explanation of the acceleration mechanism of DMCMC, we refer the reader to Appendix E.1.
>
> **Implementation**
>
> DMCMC does not resemble PC even from the perspective of implementation. MCMC in PC runs during reverse-S/ODE integration, and MCMC in DMCMC runs before reverse-S/ODE integration:
>
> * PC : (prior noise) &rarr; (integ. step) &rarr; (MCMC) &rarr; (integ. step) &rarr; (MCMC) &rarr; $\ldots$ &rarr; (clean data sample)
> * DMCMC : (MCMC to generate points near image manifold) &rarr; (integration) &rarr; (clean data sample)
>
> Moreover, as already mentioned, MCMC in PC runs in $\mathcal{X}$ at a fixed $\sigma$ whereas MCMC in DMCMC runs in $\mathcal{X} \times \mathcal{S}$ by adaptively updating $\sigma$.
>
> **Empirical**
>
> Finally, DMCMC can also be used to accelerate PC algorithms as well, as shown in the table below. This means MCMC in DMCMC and PC play orthogonal roles in generating clean data. If DMCMC were a straightforward extension of PC, we would not see such acceleration. This table is added as Figure 13 in Appendix D.1 in the revised version of our paper.
>
> *Review Table 1: PC acceleration of DLG on CIFAR10 (FID).*
>
> | Method | NFE 50 | NFE 100 | NFE 200 | NFE 400 |
> |---|---|---|---|---|
> | PC | 475.07 | 373.46 | 32.21 | 3.7 |
> | PC+DLG | 249.62 | 47.85 | 7.74 | 2.66 |
>
> **References**
>
> [1] Score-Based Generative Modeling through Stochastic Differential Equations, ICLR, 2021.

---

> ### Author Response · Authors · 2022-11-16
> **Do our revisions and responses answer your concerns and questions?**
>
> Dear Reviewer 7H8V,
>
> We thank the Reviewer for the constructive comments. As the end of the paper revision period is approaching, we would like to ask whether our paper revisions and responses have addressed your concerns and questions adequately. If not, we would be happy to discuss and update our paper further.
>
> Regards,
>
> the authors.

---

> > ### Comment · Reviewer_7H8V · 2022-11-26
> > **thanks for the reply**
> >
> > thanks for the response and more experimental support! I think the paper quality has improved with more experimental results. I've increased my score.

---

> > > ### Author Response · Authors · 2022-11-27
> > > **Thank you for raising the score!**
> > >
> > > Thank you for reconsidering our paper and raising the score! We are certain your comments have helped us clarify the contributions of our paper and solidify the experiment results.

---

### Author Response · Authors · 2022-11-12
**General Reply to All Reviewers**

We sincerely thank all the Reviewers for their valuable comments. We have tried our best to incorporate all suggestions, comments, and questions into the revised version of our paper. The modified parts are highlighted in blue. In our paper, we have made the following revisions:

**Abstract and Introduction** We reworded some possibly misleading sentences. (Please note that we could not modify the abstract displayed on the OpenReview page for our paper.)

**Section 3.2** Clarified how DMCMC accelerates sampling.

**Section 5.2 Table 1** Added performance comparison with other fast solvers.

**Section 7** Added reproducibility statement containing an anonymous download link to our main experiment codes.

**Appendix C** Added pseudocodes for DLG and DLG starting point generation.

**Appendix D** Added more related works.
* **Appendix D.1** Differences between DMCMC and the Predictor-Corrector sampler.
* **Appendix D.2** Differences between DMCMC and previous works that use a noise predictor network.

**Appendix E** Added more discussions.
* **Appendix E.1** More explanation on how DMCMC accelerates sampling.
* **Appendix E.2** Trade-off between the convergence speed of MCMC and sharpness of prior $\hat{p}(\sigma)$.
* **Appendix E.3** VE Diffusion vs. VP Diffusion in DMCMC.

**Appendix F** Added more experiments.
* **Appendix F.1** Evolution of $\sigma$ during Langevin Gibbs on $\mathcal{X} \times \mathcal{S}$.
* **Appendix F.2** Denoising step ablation.

---

### Decision · Program_Chairs · 2023-01-20

**Decision:**

Reject

**Justification For Why Not Higher Score:**

The negatives are that the paper makes specific choices, such as using the variance exploding SDE, so it's unclear if the proposed method would generalize to other types of diffusions that often yield better results, such as the critically damped langevin difffusion. As noted in the author reply, not considering other diffusions seems to limit other techniques to speed up sampling. The other negatives are that the writing as evidenced by details queried by requested of the reviewers and that the prior on the variances ties the overall mixing of the gibbs sampler with the rate at which samples get accepted (when the noise is low).

**Justification For Why Not Lower Score:**

N/A

**Metareview: Summary, Strengths And Weaknesses:**

The paper develops a different way to sample from a diffusion-based generative model. Diffusion based generative modeling typically transforms noise to something that looks like a data by simulating an SDE. This simulation can be slow. To make generation faster the authors propose to augment the sampling space with variances or times and generate on the augmented space using Gibbs sampling. The idea is that favoring smaller times will keep samples closer to data rather than having to start from scratch.

The positives are sampling from a diffusion faster is a very important problem and the idea of augmenting the space to sample the noise is fresh; the samples evaluate well with FID.

The negatives are that the paper makes specific choices, such as using the variance exploding SDE, so it's unclear if the proposed method would generalize to other types of diffusions (reviewer uZQw) that often yield better results, such as the critically damped langevin difffusion. As noted in the author reply, not considering other diffusions seems to limit other techniques to speed up sampling. The other negatives are that the writing as evidenced by details queried by requested of the reviewers and that the prior on the variances ties the overall mixing of the gibbs sampler with the rate at which samples get accepted (when the noise is low) (reviewer DAST).

The author reply and revision cleans up some of these issues (one of the reviewers increased their score), but the majority of the changes are relegated to the appendix. The work would be stronger if some more of this material was weaved into the main text.